# Addressing the dichotomy of fishing and climate in fishery management with the FishClim model

Grégory Beaugrand [1✉], Alexis Balembois [1], Loïck Kléparski[1,2] & Richard R. Kirby [3,4]

The relative influence of fishing and Climate-Induced Environmental Change (CIEC) on long-term fluctuations in exploited fish stocks has been controversial[1-3] because separating their contributions is difficult for two reasons. Firstly, there is in general, no estimation of CIEC for a pre-fishing period and secondly, the assessment of the effects of fishing on stocks has taken place at the same time as CIEC[4]. Here, we describe a new model we have called FishClim that we apply to North Sea cod from 1963 to 2019 to estimate how fishing and CIEC interact and how they both may affect stocks in the future (2020-2100) using CMIP6 scenarios[5]. The FishClim model shows that both fishing and CIEC are intertwined and can either act synergistically (e.g. the 2000-2007 collapse) or antagonistically (e.g. second phase of the gadoid outburst). Failure to monitor CIEC, so that fisheries management immediately adjusts fishing effort in response to environmentally-driven shifts in stock productivity, will therefore create a deleterious response lag that may cause the stock to collapse. We found that during 1963-2019, although the effect of fishing and CIEC drivers fluctuated annually, the pooled influence of fishing and CIEC on the North Sea cod stock was nearly equal at ~55 and ~45%, respectively. Consequently, the application of FishClim, which quantifies precisely the respective influence of fishing and climate, will help to develop better strategies for sustainable, long-term, fish stock management.

[1] Univ. Littoral Côte d'Opale, CNRS, Univ, Lille, UMR 8187 LOG, F-62930 Wimereux, France. [2] Marine Biological Association, Citadel Hill, Plymouth PL1 2PB, UK. [3] The Secchi Disk Foundation, Kiln Cottage, Gnaton, Yealmpton, Devon PL8 2HU, UK. [4] Ronin Institute, Montclair, NJ 07043, USA. ✉email: gregory.beaugrand@univ-lille.fr

Managing fish stocks has always been a difficult task because stocks exist in complex ecosystems that can experience substantial changes triggered by extrinsic (e.g. fishing and CIEC, see definition of CIEC in Table 1) and intrinsic (e.g. biological or ecological processes) forces[3,6,7]. These changes can result in stock collapse due to overexploitation[7–11] or climate-induced alterations in spatial range with consequences upon local fish abundance[12–14]. Although many studies have investigated how fishing and environment may interact to affect a fish stock[15–18], the precise respective contribution of fishing and CIEC and how this varies in time remains poorly known, yet this knowledge is likely to be fundamental to effective fisheries management[19,20].

The Atlantic cod *Gadus morhua* L. has declined in the North Sea since the end of the gadoid outburst[21] and there has been a debate on whether or not CIEC has contributed with over-fishing to the diminishing Spawning Stock Biomass (SSB)[1,6,22,23]. Surprisingly, although some studies have jointly investigated the influence of CIEC and fishing on cod SSB[6,15,24], there have been no attempts to quantify precisely the effects of the two drivers despite their importance in terms of stock management. As a result, current management practices continue to ignore the potential influence of CIEC on cod stocks[24]. This is especially worrying since anthropogenic climate change is having a discernible influence on many marine ecosystems and that its impacts may drastically increase in the decades to come[25–29].

To investigate the influence of fishing and CIEC and how they might interact to affect the North Sea cod stock, we designed a model where the size of cod population (standardised Spawning Stock Biomass or dSSB hereafter, see Table 1 for a list of acronyms) depended upon (i) population growth rate $r$, (ii) fishing intensity $\alpha$ and (iii) maximum standardised SSB (called mdSSB hereafter) that can be reached in space and time and can only result from CIEC in the absence of exploitation ("Methods"). We have called this model FishClim and we applied it to the north-east Atlantic (seas around the UK) at a spatial resolution of 0.25° latitude × 0.25° longitude, with an emphasis on the North Sea cod stock.

## Results

**Spatial changes in maximum standardised SSB**. Using "Fish-Clim", we modelled the spatial patterns in maximum standardised Spawning Stock Biomass for 1997–2019, called hereafter mdSSB (i.e. depending only upon the environment, no fishing). mdSSB was, reassuringly, close to our knowledge of the spatial distribution of cod in the north-east Atlantic (Fig. 1a)[30–32].

**Temporal changes in maximum standardised SSB**. We then assessed average long-term changes in mdSSB in the North Sea (51°N–62°N and 3°W–9.5°E). We found a good correlation between long-term changes in mdSSB and recruitment at age 1 with a 1-year lag (Fig. 1b, correlation $\sigma = 0.79$, probability corrected for autocorrelation $p_{ACF} = 0.02$, $n = 56$ years). In addition to be expected biologically because recruitment is assessed at age 1, the 1-year lag was also found in some studies that investigated relationships between changes in plankton and cod recruitment[6,33]. Long-term changes in mdSSB were also highly correlated with long-term changes in a plankton index updated for the period 1958–2017 with no lag ($\sigma = 0.73$, $p_{ACF} = 0.04$, $n = 60$ years, Fig. 1c). These results are interesting because they show that our model reflects well the trophic environment of cod at the larval stage[33] and probably integrates natural mortality well, which is greatest at age $\leq 1$[34]. The correlation was not significant for ICES SSB (with or without a lag) because changes in SSB are strongly influenced by fishing, a driver that was not considered in this first analysis ($\sigma = 0.52$ and $p_{ACF} = 0.23$ for both correlations, $n = 57$ and 56 years for no lag and a 1-year lag, respectively, Fig. 1d). This result shows that CIEC cannot by itself

**Table 1 List of acronyms and main symbols used in the text. Other symbols can be found in the "Methods" section.**

| Acronym/symbol | Meaning | Definition |
|---|---|---|
| CIEC | Climate-Induced Environmental Changes | All environmental alterations that result from climatic variability and anthropogenic climate change. In this paper, we considered changes in sea surface temperature, chlorophyll-a concentration and a sliding 15-day period above a chlorophyll-a concentration level of 0.05 mg.m$^{-3}$. |
| SSB | Spawning Stock Biomass | Total weight of a fish stock able to reproduce. |
| dSSB | Standardised SSB | SSB standardised between 0 and 1. |
| mdSSB | Maximum standardised SSB | Maximum dSSB in the absence of fishing. Only the environment influences mdSSB in space and time. mdSSB varied between 0 (unsuitable environment) and 1 (perfectly suitable environment). |
| ICES SSB | SSB from ICES, expressed in decimal logarithm | See ICES[35] |
| ICES dSSB | Standardised ICES SSB | SSB data from ICES[35] standardised in a way to include it in the FishClim model (Supplementary Fig. 3 and "Methods"). |
| r | Population growth rate | See Eq. (1) in "Methods" |
| α | Fishing intensity | See Eq. (1) in "Methods" |
| K | mdSSB | See Eq. (1) in "Methods" |
| σ | Coefficient of linear correlation | See Sokal and Rohlf[97] |
| $p_{ACF}$ | Probability after accounting for temporal autocorrelation | See Pyper and Peterman[91] |
| n | Number of years used in the calculation of correlations | – |
| F | Fishing effort | See ICES[35] |
| ESM | Earth System Model | See Supplementary Text 1 |
| MSY | Maximum Sustainable Yield | The fishing effort that allows the maximum number of fish to be harvested over the long-term without a decline in the stock |
| SSP245 | Shared Socio-economic Pathways 245 | "Middle of the road" scenario |
| SSP585 | Shared Socio-economic Pathways 585 | "Fossil-fueled development" scenario |

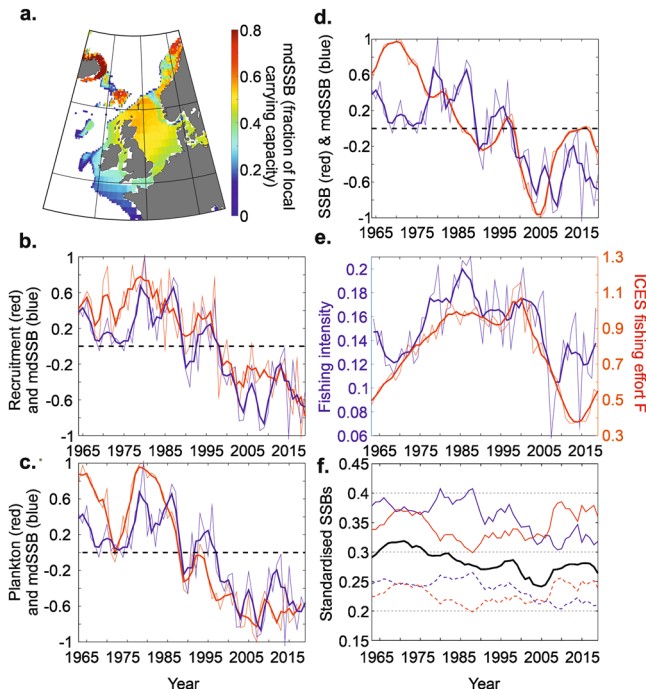

**Fig. 1 Maximum standardised spawning stock biomass mdSSB (K) and fishing intensity (α) modelled by FishClim in relation to observed changes in a plankton index of larval cod survival, recruitment at age 1, Spawning Stock Biomass (SSB) and ICES fishing effort F in the North Sea. a** Spatial patterns of average mdSSB (i.e. without fishing) for the period 1997–2019 (i.e. measured chlorophyll data). **b** Long-term changes in cod recruitment at age 1 with a lag of 1 year (red) in relation to long-term changes in mdSSB (blue); **c** Long-term changes in mdSSB (blue) in relation to long-term changes in a plankton index of larval cod survival (red) updated from Beaugrand and colleagues[33]. Long-term changes in mdSSB (1963-2019) were based on modelled daily chlorophyll data. **d** Long-term changes in cod ICES SSB with a lag of 1 year (red) in relation to long-term changes in mdSSB (blue). **e** Long-term changes in estimated fishing intensity α (blue) in relation to long-term changes in ICES fishing effort F (red). All time series in (**b**–**d**) were standardised between −1 and 1 and thick lines in (**b**–**e**) were the original time series smoothed by means of a first-order simple moving average. **f** Modelled standardised SSB based on long-term changes in the environment and assessed fishing intensity (thick black line) with (i) modelled standardised SSB based on a constant environment fixed to the minimum (dashed red curve) or optimal (full red curve) value observed for 1963–2019 and long-term changes in fishing intensity α and (ii) modelled standardised SSB based on a constant fishing intensity fixed to the mimimum (full blue curve) or the maximum value (dashed blue curve) observed for 1963–2019 and long-term climate-induced environmental changes.

explain long-term fluctuations in North Sea cod SSB although it well explains recruitment at age 1.

**Assessing fishing intensity in 1963–2019.** Using North Sea ICES SSB that we included in Eq. (10) ("Methods" and Supplementary Fig. 3), we assessed fishing intensity α for 1963–2019. Long-term changes in our estimates of fishing intensity α were positively correlated ($r = 0.56$, $P_{ACF} = 0.04$, $n = 56$) with ICES fishing effort F[35] (Fig. 1e). The estimation of α allowed us to reconstruct long-term changes in cod ICES dSSB and to examine the respective influence of fishing and CIEC by means of Eq. (1) ("Methods") using four hypothetical scenarios ("Methods", Fig. 1f): (i–ii) constant minimum and maximum fishing of 1963–2019 and

year-to-year CIEC and (iii–iv) constant minimum and maximum CIEC of 1963–2019 and year-to-year changes in fishing intensity α. Our model predicted lowest dSSB when mdSSB ($K_t$ in Eq. (1)) corresponded to an unsuitable CIEC or when fishing intensity was high (dashed blue and red curves in Fig. 1f). The opposite conditions, a favourable CIEC and low fishing intensity, led to highest dSSB (full blue and red curves in Fig. 1f).

Our results therefore show clearly, how fishing and environment interact to influence a stock (Fig. 1f). For example, if environmental conditions remain suitable, as they were during the Gadoid Outburst (~1963–1983)[33,36], the reduced fishing pressure from the end of 2010 onwards would have led to a new outburst in cod even more prominent than observed between 1963–1983 (full red curve in Fig. 1f). Further, if the level of fishing intensity was constantly the lowest observed during the time period, the FishClim model suggests that dSSB observed during the second phase of the gadoid outburst would have been much higher (full blue curve). Long-term changes in reconstructed (ICES) dSSBs (thick black curve) shifted from being closer to the upper (full red and blue) curves (i.e. suitable environmental conditions or low fishing intensity) during the Gadoid Outburst to being closer to the lower curves (less suitable environmental conditions or high fishing intensity, dashed red and blue in Fig. 1f), which suggests that either fishing or climate, or both, have negatively affected cod dSSB.

**Identification of the influence of fishing and climate/environment on spawning stock biomass.** We examined the respective influence of fishing and CIEC on ICES SSB during different time periods (P1-P7) of 1963–2019 as revealed by a cluster analysis performed on long-term reconstructed changes in (ICES) SSB, fishing and CIEC influences (Fig. 2, "Methods"). The highest ICES SSB, which was observed during the first period of the Gadoid Outburst (time period P2 in Fig. 2a) was the result of a positive environmental influence (including favourable plankton), at a time of moderate fishing intensity (Fig. 2a, b). Despite an increase in the environmental influence and its positive effect on cod recruitment during the second phase of the Gadoid Outburst (*circa* time period P3, Fig. 2a, see also Fig. 1a, b), SSB diminished strongly because of an increase in fishing intensity that strengthened further until the end of the 1980s (Fig. 2a, b). From the end of the 1980s to 2007 (~P4–P6), the pronounced reduction in SSB paralleled rapid, adverse changes in environmental suitability that negatively affected recruitment when there was also considerable fishing effort. This led to a period (2000–2007, P6) of lowest SSB where fishing was too pronounced at a time of unsuitable environmental conditions. From 2008 onwards (P7), fishing was reduced by management[35] and as a result SSB increased despite an environment that remained highly unsuitable for recruitment (Figs. 1a, b & 2a, b). These results show that both fishing and CIEC affected North Sea cod SSB.

**Quantification of the influence of fishing and climate/environment on spawning stock biomass.** To quantify the influence of fishing and CIEC on (ICES) SSB, we calculated an index of fishing influence (expressed in percentage, "Methods"). Overall for the period 1963–2019, using a resampling procedure (i.e. Jackknife, "Methods"), we found that changes in fishing intensity and in CIEC were 55% (range between 55% and 56%) and 45% (range between 44% and 46%) respectively, suggesting that both drivers contributed almost equally to the long-term changes in cod SSB in the North Sea. A global estimation masks important temporal changes in the varying temporal influence of fishing and CIEC, however (Fig. 2c). During the first period of the Gadoid Outburst (P2, Fig. 2), the two drivers contributed almost equally

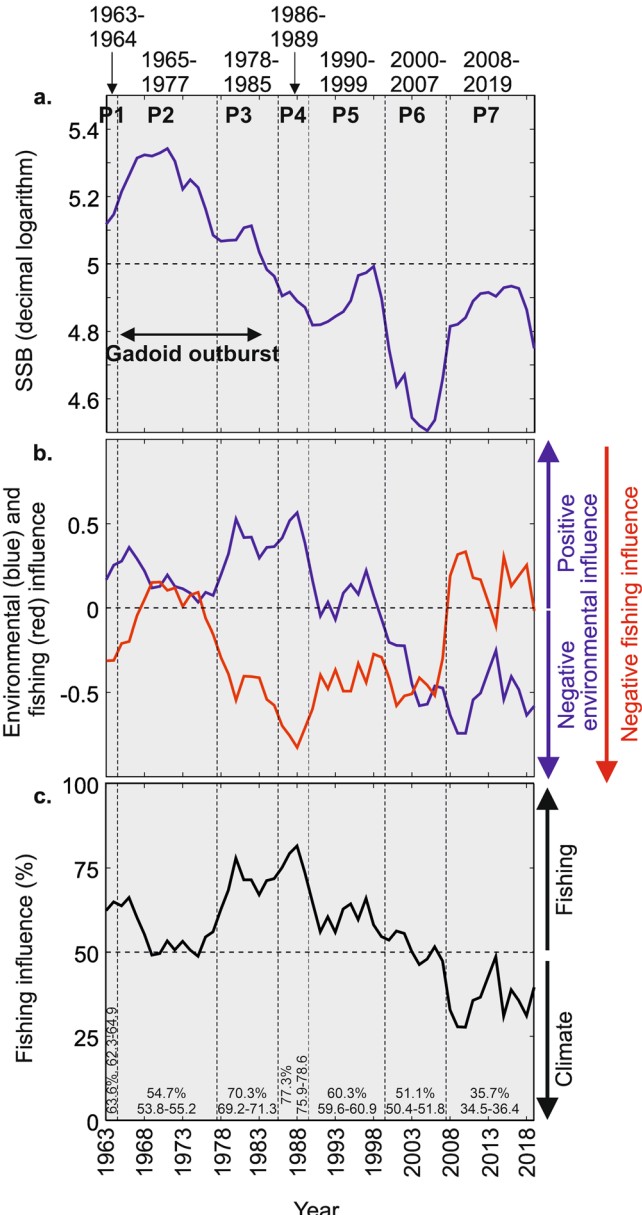

**Fig. 2 Respective contribution of the influence of the environment and fishing intensity on Spawning Stock Biomass (SSB) for 1963-2019.**
**a** Long-term changes in ICES SSB (decimal logarithm). The timing of the Gadoid Outburst is indicated. **b** Long-term changes in the estimated positive environmental (blue) and negative fishing intensity (red) influence on SSB. **c** Long-term quantification of the fishing/environmental influence on SSB. Dashed black vertical lines denote the different time periods P1-P7 identified by the cluster analysis based on the time series shown in (**a**, **b**). A quantification of the influence of fishing (in percentage) is indicated at the bottom of panel **c** for each time period (average, minimum and maximum values) after applying a jackknife procedure.

(Fig. 2c, fishing influence was ~55%). During the second phase of the Gadoid Outburst and until the end of the 1980s (P3–P4), the influence of fishing predominated (between ~69% and ~78% on average). Then, the influence of fishing rapidly diminished *circa* 1990 and stabilised between ~59% and ~61% on average until the end of the 1990s (P5), coinciding with a pronounced, adverse environmental shift that ushered in sustained, adverse environmental conditions. A pronounced fishing intensity associated with the regime change triggered a rapid collapse of cod SSB in

2000–2007 (P6); the contribution of fishing and CIEC was equal (between 50 and 51%). A reduction in fishing intensity due to fish management allowed the stock to avoid collapse and fishing effort reduced to reach a value of ~34–36% from 2008 (P7); this last result suggests that the current CIEC regime is strongly affecting cod SSB (~64–66%). To summarise, our analysis demonstrates how both fishing and CIEC interplayed historically to affect the current state of cod SSB in the North Sea.

**Understanding how fishing and climate/environment interact presently, and in the future.** Climate change (natural and/or anthropogenic) has affected the environment of the North Sea by altering plankton composition and ecosystem trophodynamics[33,37,38]. We forced our model by outputs from four Earth System models (ESMs) based on two scenarios of SST/Chlorophyll changes (i.e. Shared Socio-economic Pathways SSP245 and SSP585, "Methods") to assess mdSSB ($K_t$ in Eq. (1)) for the period 1850–2100 and examined the potential influence of anthropogenic climate change. Although our estimates showed pronounced inter-ESM variability for both emission scenarios (i.e. thin black curves and average in thick green for 1850–2019, thin dashed blue and red curves for 2020–2100 for scenarios SSP245 and SSP585, respectively), future mdSSB (i.e. with no fishing) were predicted to decrease substantially during the forthcoming century (Fig. 3a, thick full blue and red curves for the average of all SSP245 and SSP585 scenarios, respectively). Differences in mdSSB due to the magnitude of anthropogenic climate change (i.e. the difference between the average of the four scenarios SSP245 and SSP585) reinforced from ~2050 and reached an average of 0.09 in term of mdSSB, with a range of 0.08–0.13, for the last decade of the 21st century, a reduction of 36.1% of mdSSB (range of 30.8–43.8% when based on all individual years of the last decade). Adding a constant (standardised) catch, corresponding to the average of 2008–2019 (P7, i.e. $\alpha X = 0.03$ in Eq. (1)), to the "middle of the road" scenario (i.e. SSP245) led to a reduction of dSSB of about the same amplitude as the difference induced by warming intensity (Fig. 3a, thick dashed versus full blue curves); i.e. a reduction of 39% (range of 33.3–44.5%). Combining the "fossil-fueled development" scenario with a constant catch (using the same value as above) led to a pronounced stock reduction from 2082 to 2087, followed by full extirpation (dSSB = 0) from 2088 onwards (Fig. 3a, thick dashed red line).

To understand how fishing and the environment interact we estimated dSSB as a function of both fishing intensity and CIEC including superimposed long-term changes in (ICES) dSSB in the North Sea (1963–2019; Fig. 3b, c, "Methods"). mdSSB (ordinate on Fig. 3b) denotes the maximum dSSB achievable for a given environmental regime; i.e. dSSB is always below mdSSB. Expectedly, alleviating fishing effort is the only way to maintain a stable SSB when the environmental regime becomes less suitable[39]. Although it is possible to maintain cod SSB when the environment is highly suitable, such as the Icelandic cod stocks for the current CIEC regime (e.g. for $K_t > 0.5$), it is harder, if possible, to achieve in the environmentally less favourable North Sea (Fig. 3b and Fig. 1a). This can be illustrated by the three points A, B and C in Fig. 3b. For a hypothetical dSSB corresponding to point A, we see that increasing dSSB by fish management (i.e. along the horizontal line from the starting point A to the left on the figure) is easier than for a dSSB corresponding to points B and C (Fig. 3b); this is because the number of isolines to the left of each point, reflecting the scope to reduce fishing intensity, decreases from A to C. At point C, it becomes nearly impossible to keep dSSB stable by cod management because the number of isolines is considerably reduced along the horizontal

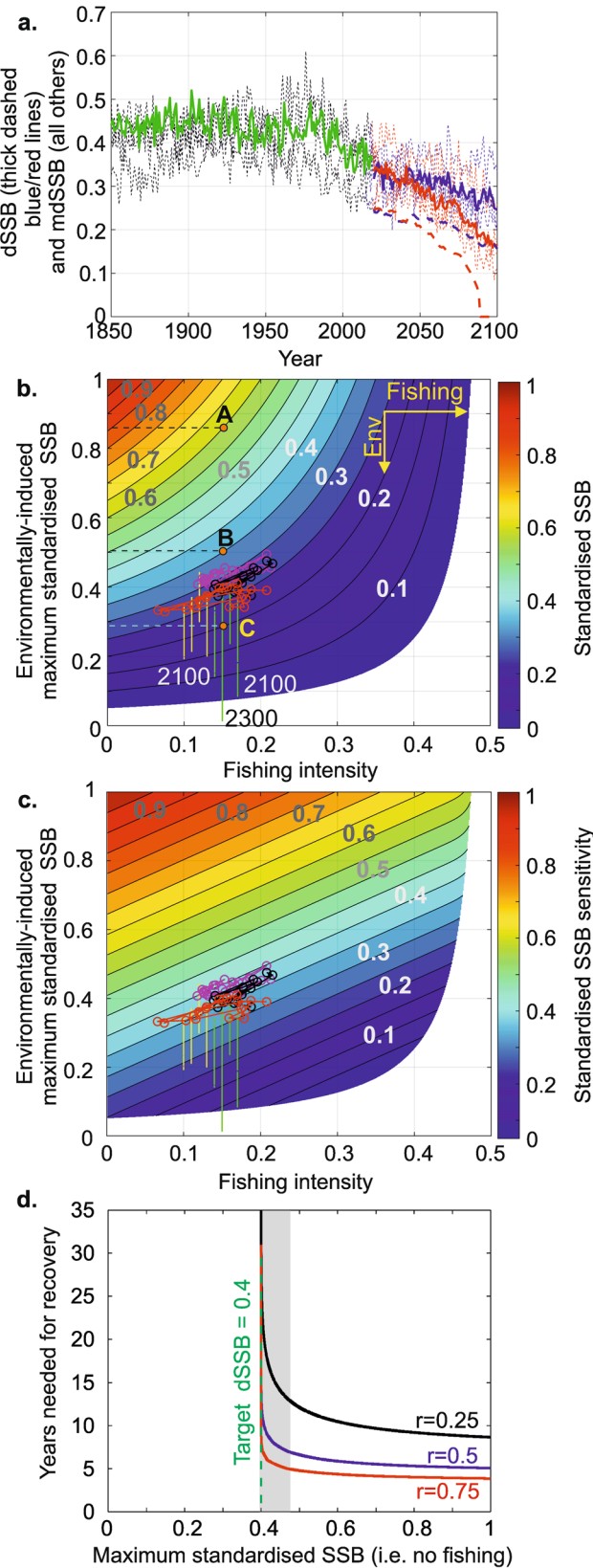

**Fig. 3 Long-term changes in Spawning Stock Biomass for 1850–2100 and interactive influence of the environment and fishing. a** Long-term changes in maximum standardised SSB (dashed black thin lines and full thick green, red and blue lines) and standardised SSB (red and blue dashed thick lines) for 1850–2100. The thick full green line is the average of mdSSB based on four ESMs (Earth System Models; four dashed thin black curves) for 1850–2019. The thick full blue and red lines for 2020–2100 are the average of the four estimates (one for each ESM) based on scenarios SSP245 (the four dashed thin blue lines) and SSP585 (the four dashed thin red lines), respectively. The dashed thick blue and red lines are trajectories based on a constant standardised catch, averaged for the last period 2008–2019 identified by a cluster analysis, with scenarios SSP245 and 585, respectively. **b** Standardised SSB as a function of maximum standardised SSB (i.e. environmental influence only) and fishing intensity. The three brown points A, B and C are three hypothetical levels of dSSB. Env: environment. **c** Sensitivity of standardised SBB to the environment and fishing. In (**b**, **c**), circles are standardised ICES SSB based on years from 1963 to 2019 (magenta: 1963–1985, black: 1986–1999, and red: 2000–2019). Yellow and green dots are standardised SSB for 2020–2100 (or 2300 exclusively for Scenario SSP585 of IPSL ESM) based on four ESMs and scenarios SSP245 and 585, respectively. Fishing intensity, unpredictable for 2020–2100, was fixed to be arbitrarily constant between 0.08 and 0.17 by increment of 0.1 for display purpose (i.e. high resolution of the colour diagram), starting by ESMs based on scenario SSP245 followed by scenario SSP585. **d** Number of years needed for recovery of the stock to a target standardised SSB (dSSB) of 0.4 (vertical dashed green vertical line) after stock collapse for three different population growth rates: 0.25 (black), 0.5 (blue) and 0.75 (red). The grey zone denotes an area where recovery slows down when the maximum standardised SSB (mdSSB) approaches the target dSSB; such a situation occurs when the environment becomes less suitable. No fishing is allowed here (i.e. a hypothetical moratorium).

influence a stock negatively[39]. However, our results suggest that in the context of anthropogenic warming manage the stock by reducing fishing effort alone will reach a limit as the stock diminishes as a consequence of CIEC. Some of our scenarios even forecast a collapse either in 2100 (UKESM1 model, SSP585) or 2300 (IPSL model, SSP585, Fig. 3c).

We investigated theoretically how many years it would take to recover to a dSSB of 0.4 (close to the current average, see Fig. 1a) after a hypothetical collapse of the North Sea cod stock (i.e. dSSB = 0.1 in Eq. (1), "Methods"). We assumed the rapid establishment of a fishing moratorium (i.e. fishing intensity α = 0) after such a breakdown, as was implemented when Newfoundland cod stocks collapsed[40]. Calculations were made by applying Eq. (1) ("Methods") for three values of population growth rate ($r$ = 0.25, 0.5 and 0.75). We found the stock rebuilt relatively rapidly when the environmental regime was suitable, at mdSSB = 1 from 3.9 to 5.1 and 8.6 years for $r$ = 0.75, 0.5 and 0.25, respectively (Fig. 3d). However, when conditions became less suitable and mdSSB approached the target dSSB (here, dSSB = 0.4), the stock took much more time to recover to a level suitable for exploitation, it took 8.0, 12.9 and 27.2 years for $r$ = 0.75, 0.5 and 0.25, respectively, at mdSSB = 0.401 (Fig. 3d). In other words, when mdSSB > dSSB the stock rebuilds and when mdSSB ≤ dSSB this becomes impossible.

**Potential consequences of fisheries management and climate-induced environmental changes.** We examined how fishing and CIEC may affect cod stocks and their exploitations around UK with a focus on the North Sea ("Methods"). We started by assessing year of cod extirpation for two scenarios of CIEC and two scenarios of

line from the starting point C to the left. This is well shown by an analysis of the sensitivity of dSSB as a function of mdSSB and fishing (Fig. 3c). Sensitivity of dSSB to fishing (and therefore to fish management), as well as CIEC, diminishes when dSSB decreases. Rightly, it is common practice to recommend a reduction in fishing effort when both climate and fishing pressure

cod management (constant in space and time—no adjustment—versus adjusted fishing intensity using a Management Sustainable Yield—MSY—approach to account for CIEC, "Methods"). The resulting analysis revealed that controlling fishing intensity (or fishing effort *sensu* ICES, for example) delayed cod extirpation, and this is especially true when anthropogenic climate change is strong (Fig. 4, b versus Fig. 4d, e, Fig. 4g, h); for the North Sea area we found a delay of 3 (median) and 25 years of cod extirpation between constant and adjusted fishing to account for CIEC for SSP245 and SSP585 (Fig. 4g, h), respectively. Similarly, the influence of warming was more prominent when fishing intensity was constant than adjusted in space and time to account for CIEC (Fig. 4a–c versus Fig. 4d–f); a delay of 16 and 4 years between scenarios SSP245 and SSP585 was found for constant and adjusted fishing, respectively (Fig. 4c, f). The combination of uncontrolled climate change and fishing (Fig. 4b) led to a much more rapid extirpation of cod, with delay of 28 years of cod extirpation between SSP585 associated with constant fishing and SSP245 associated with fishing adjusted to account for CIEC (Fig. 4i). Although fishing intensity was hypothetical in our scenarios of changes, the analysis clearly suggests that both drivers are important to consider in future projections.

We then assessed pooled standardised catch by 2100 (2020–2100) for two scenarios of CIEC (SSP245 and 585) and the two scenarios of cod management (constant versus adjusted—MSY—fishing intensity, "Methods"). We found that controlling fishing intensity and the magnitude of anthropogenic climate change had a strong influence on cod exploitation (Fig. 5). Not adjusting fishing intensity to account for CIEC (Fig. 5a, d versus adjusted in Fig. 5b, e) reduced pooled long-term standardised catch (2020–2100) by 9.9% (median) and 27.1% in scenarios SSP 245 and 585, respectively (Fig. 5c, f). Limiting warming (SSP 245—Fig. 5a, b—versus SSP 585—Fig. 5d, e) had a positive influence on the long-term catches as well (Fig. 5a, d, g versus Fig. 5b, e, h); a reduction in pooled standardised catch of 27.7% (median value) was observed in the North Sea when fishing was constant in space and time whereas a reduction of 12.7% (median value) was found when fishing was adjusted to account for CIEC (Fig. 5g, h). The combination of poor fish management and intense warming led to a pronounced reduction in pooled standardised catch for the whole century (Fig. 5i) with a median value of 35.8% of reduction in pooled standardised catch. In this case, we might ask, *what course of action could sustain stocks*? We suggest that mitigating anthropogenic climate change will be much more challenging[41,42] than opting for rigorous regional fish management, although both would be clearly desirable.

## Discussion

To the best of our knowledge, a few studies have examined the joint influence of climate change and fishing on cod[18,30,43]. Engelhard and colleagues[30] have investigated the influence of both drivers on the spatial distribution of cod in the North Sea over the past 100 years. The authors showed that the deepening and northward shift of cod were attributable to warming whereas the eastward shift was best explained by fishing that strongly depleted the stock off the coasts of England and Scotland. Their

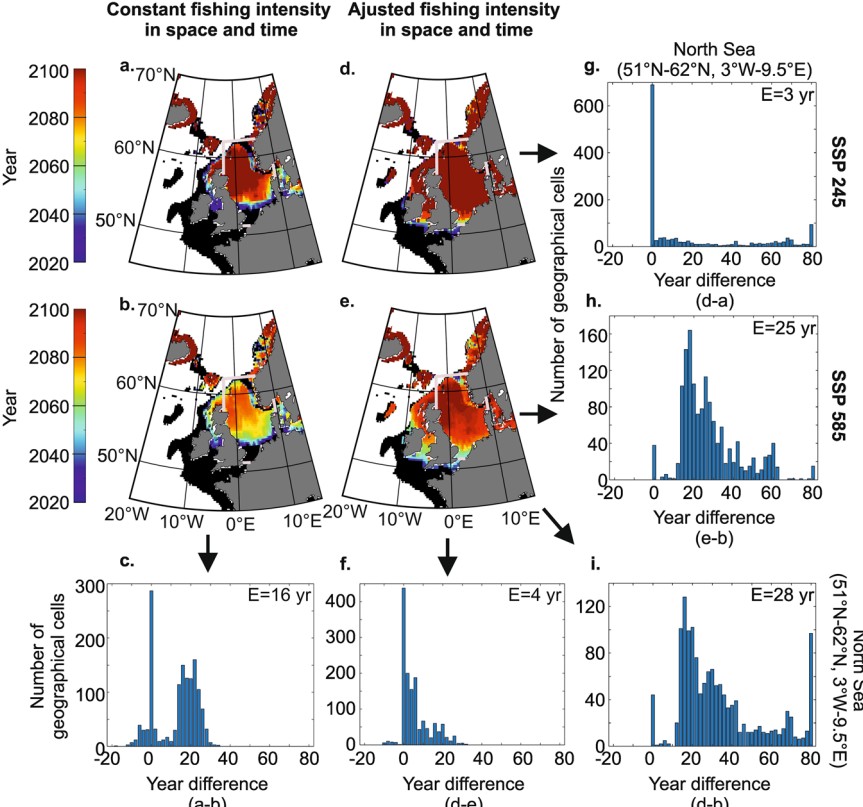

**Fig. 4 Effect of cod management and anthropogenic climate change on year of cod extirpation in 2020–2100 with a focus in the North Sea. a–d** Maps of year of cod extirpation based on a constant (**a**, **b**) and an adjusted (MSY) (**d**, **e**) fishing intensity in space and time and scenario SSP245 (**a**, **d**) and 585 (**b**, **e**). Thick magenta lines display the North Sea boundaries used to calculate histograms. **c**, **f–i** Frequency histograms of difference between maps of time to extirpation for the North Sea (51°N-62°N and 3°W-9.5°E). **c** Year difference between the maps of (**a**, **b**). **f** Year difference between the maps of (**d**, **e**). **g** Year difference between the maps of (**d**, **a**). **h** Year difference between the maps of (**e**, **b**). **i** Year difference between the maps of (**d**, **b**). The value of median *E* (expressed in year, yr) is indicated on all histograms.

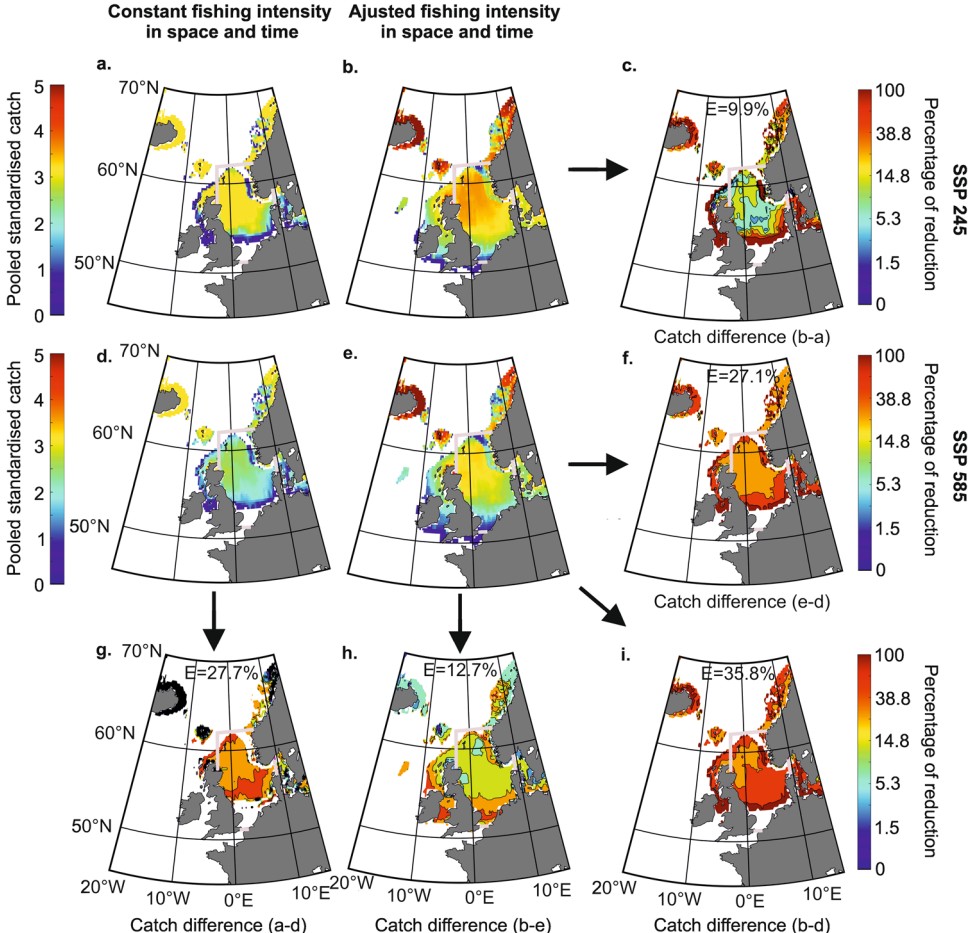

**Fig. 5 Effect of cod management and anthropogenic climate change on pooled standardised catch in 2100 with a focus in the North Sea.** Maps of pooled standardised catch (2020–2100) based on a constant (**a**, **d**) and an adjusted (MSY) (**b**, **e**) fishing intensity in space and time and scenario SSP245 (**a**, **b**) and 585 (**d**, **e**). **c**, **f–i** Maps of diminution in pooled standardised catch based on difference between maps of **b**, **a** (**c**), **e** and **d** (**f**), **a** and **d** (**g**), **b** and **e** (**h**) and **b** and **d** (**i**). Thick magenta lines display the North Sea boundaries used to calculate the median on percentage of catch diminution maps (**c**, **f–i**); the value of median E (expressed in percentage) is indicated on these maps.

study revealed the fundamental importance of both climate change and fishing pressure for our understanding of North Sea cod. In the Baltic Sea, a study investigated changes through time of the respective influence of CIEC, predation, eutrophication and exploitation on cod biomass during the 20th century and quantified their respective influence[18]. At the beginning of the 20th century, nutrient availability and mammal predation were the main drivers of the size of the stock. Then from the 1940s, fishing became dominant. In the 1980s, eutrophication starts to play a role. For the period 1980–1984, the authors assessed that the relative influence of eutrophication, climate and fishing on Baltic cod biomass was 13%, 43% and 52%, respectively[18]. Although we did not assess the potential influence of eutrophication on North Sea cod because this sea is only marginally influenced by this environmental issue[44], our estimates for fishing and climate at about the same period were >70% and <30% for fishing and CIEC in the North Sea, respectively (Fig. 2c).

Our results provide a general framework against which (i) we can better understand the respective influence of fishing and CIEC (e.g. their synergy and antagonism) on past changes in cod SSB and (ii) how to anticipate and mitigate future changes by adjusting fishing intensity. Climate change affects recruitment by diminishing larval cod survival[6,33], a process that takes place in the upper water column through the direct influence of temperature on physiology and its indirect effects through plankton

composition[6,33,45]. Although any changes in the recruitment affect subsequently SSB, fishing affects it directly, which in turn increases the sensitivity of the species to climate change though many processes (e.g. maternal effect, migration, demographic structure)[46–48].

Our study provides evidence that both fishing and CIEC interacted to affect long-term changes in cod SSB in the North Sea (Fig. 2), sometimes acting either synergistically (e.g. collapse of SSB from the end of the 1980s to 2005, P5–P6) or antagonistically (e.g. P2, P3 and P7). Our results therefore emphasise how both fishing and climate must be considered to resolve the apparent dichotomy (i.e. the debate between the respective contribution of fishing and environment on a fish stock) they create for fisheries management[1,19].

The synergistic interaction of fishing and CIEC indicates that it is critical to control fishing intensity during anthropogenic climate change if we want to exploit this wildlife sustainably as a food resource (Fig. 3). Mitigating climate change is therefore an important consideration[42] because dSSB might become so low under a "fossil-fueled development" scenario (i.e. SSP585) that cod management will be unable to prevent the environmental influence of anthropogenic climate change (Fig. 3b, c). Our results show that high warming reduces the possibility of cod management but the absence of management exacerbates the impact of warming (Figs. 3–5).

We also provide an explanation why, despite the fishing moratorium near Newfoundland, recovery, although partial, took more than two decades[49] (Fig. 3d). It is notable that recovery has proved to be very difficult for many other fish stocks (e.g. haddock, flatfish)[50]. Consequently, our results show that in the context of anthropogenic climate change, fisheries management is essential to prevent a stock collapse, up to the point where CIEC becomes so extreme that cod extirpates. In addition, our results suggest that preventing collapse is easier than trying to reverse a collapse. This is particularly true if managers try to rebuild to a level that is no longer possible under a new environmental regime[24]. These findings show how important is to manage fish stocks using dynamic reference points[51].

The FishClim model is structured in space and time, and includes both fishing and environmental effects, which make it possible to assess the respective influence of the two drivers in space and time. Although our model assesses a standardised SSB that could subsequently be scaled to the actual SSB of a stock, it does not include information on its size/age structure, which is considered to be important for management purposes[52]. In addition, our present version of the model does not include natural mortality because this process is difficult to assess with confidence at the scale of our study;[53] here we assumed it was integrated into the second term of Eq. (1) ("Methods").

The two time series of fishing intensity ($\alpha$ in our model) and effort (ICES F) were significantly correlated positively (Fig. 1e). They exhibited similar long-term patterns with a pronounced increase in fishing intensity $\alpha$ and effort F in the mid-1960s, a strong reduction in the mid-2000s and high values between these two periods. In addition, low periods of fishing intensity and effort were observed at the beginning and the end of the time period (before the mid-1960s and after the mid-2000s). The medium correlation, although significant, was mainly due to year-to-year variance in the estimations of fishing intensity/effort that might originate from the difficulty in assessing such parameters[7]. Nevertheless, given the different methods used to assess fishing intensity ($\alpha$) and effort (ICES F), we think that it is reassuring that the two time series exhibit similar long-term changes (Fig. 1e).

Although we assessed the influence of $r$ on timing for recovery after a hypothetical collapse (see Fig. 3d), we performed most analyses with a constant population growth rate $r = 0.5$. Population growth rate r is likely to be affected by temperature and food availability[54,55] but it remains strongly determined by the life history traits of a species. For example, $r$ would be higher for a $r$- than a $K$-strategy species[56]. Nevertheless, a dynamic $r$ might be easily employed in our model but it is difficult to know how temperature and chlorophyll concentration may jointly affect $r$ and in practice, it might be difficult to implement realistic changes in $r$[54].

Migration was not accounted for into the model. In this study, we assumed that (i) migration had a small influence on standardised SSB at the scale of the North Sea[57]. Although some studies have suggested that cod migration was limited to 500 km at maximum[58], more recent estimates suggest that this value was perhaps too extreme[59]. More recent studies found that in the summer (mid-June to mid-August) the range of cod movement was less than 1 km[60]. Evidence from electronic tagging experiments also suggests that there were behavioural differences between the English Channel and the North Sea cod that limit their mixing in the two areas[59]. Other mark and recapture experiments, as well as genetic evidence, have suggested that populations from the northern North Sea (>57°N) did not intermix significantly with those from the southern North Sea (<56°N)[61,62].

Although being largely debated for decades because of uncertainties on the estimates (e.g. lack of reliability and poor assumptions in some models), or because it is too specific and does not include other fisheries[63–68], we chose to use $B_{MSY}$ because it remains widely used by agencies regulating fisheries and in North Sea cod management[35,67]. However, our model can be employed with any biological reference points such as those currently discussed in the litterature[51,68].

Multispecies Maximum Sustainable Yield (MMSY) is being increasingly used[69]. The effect of multispecies fishing in our model would be to lead to an underestimate of $\alpha$. This potential issue could be partially solved by subdividing $\alpha$ into two components $\alpha_1$ (i.e. direct fishing effect) and $\alpha_2$ (indirect fishing effect). MMSY remains not easy to implement at the organisational community level because it is challenging to maximise all stocks simultaneously and inevitably there are some stocks that might be overfished while others might be underfished[67]. Nevertheless, our approach based on $B_{MSY}$ remains important because our model proposes a dynamic MSY that is adjusted as a function of environmental changes.

To estimate the maximum standardised Spawning Stock Biomass (mdSSB, $K_t$ in Eq. (1)), our FishClim model used an empirical niche model, i.e. a multiplicative empirical model that integrates temperature, bathymetry and chlorophyll-a (duration and concentration). Although the niche is composed of more ecological dimensions, the three chosen parameters are key for fish distribution[14,70]. The values of the different parameters of the niche were fixed according to our knowledge of the fish[6,23,31] and slight modifications in the values of these parameters did not alter our conclusions. Our models could be forced by any ecological niche models (or species distribution models) such as the Non-Parametric Probabilistic Ecological Niche Model (NPPEN) or the Maximum Entropy (MaxEnt) model to assess mdSSB[31,71].

Inter-ESM variability remains important and it is clear that this affects our projections (Figs. 3–5). In addition, emission scenarios are inherently unpredictable and this might also influence our projections, although in more expected ways (Figs. 4, 5). However, the model we propose could be used on a year-to-year basis to better anticipate future changes in SSB and predict more realistic fishing quotas that may either prevent stock collapse or better optimise exploitation.

## Conclusions

Forty-two years ago, McEvoy in his book The Fisherman's Problem, highlighted the dichotomy between fishing and climate that made fisheries management an intractable conundrum[19]. Although this dichotomy has waned over time and that more and more studies are considering the influence of the two drivers[15–18,72], this dichotomy has regularly reappeared since then[1,22]. Our results show that we should abandon the debate as to whether fishing is more important than CIEC;[1] simply, a stock of fish is a renewable resource the size of which is balanced by gains (recruitment and immigration) and losses (fishing, natural mortality and emigration). Both fishing and CIEC drivers have clearly influenced the North Sea cod stock, they are intricately intertwined, acting synergistically or antagonistically at different times depending upon their relative strengths (Fig. 2c). Failure to regulate fishing can have considerable adverse effects on the stock and may lead ultimately, to its collapse;[73] a breakdown in the North Sea cod stock was probably only avoided at the end of 2000s by the reduction in fishing effort after the period of strong fishing effort associated with pronounced adverse CIEC[35]. Although managing fish stocks is probably more locally achievable than mitigating global climate change on local, regional or global scales, our study also highlights the importance of limiting anthropogenic climate change as it may alter the North Sea environment in such a way that future collapses might become

unpreventable and irreparable by management. Our study also emphasises that it is likely to be particularly important to consider the position of a fishery with regard to a species environmental niche as the relative influence of CIEC will vary[23]. Although our analysis focused on North Sea cod because of the depth of understanding of this fishery and the comprehensive data available, we expect our findings to be applicable to other Atlantic cod stocks or exploited species and so we encourage a better consideration of fishing and CIEC in all future fisheries management. Failure to monitor CIEC and for fisheries management to not immediately adjust fishing effort when the environment changes will create a deleterious response lag.

## Methods
### Data
*Sea Surface temperature (1850–2019)*. Sea Surface Temperature (SST, °C) from 1850 to 2019 originated from the COBE SST2 1° × 1° gridded dataset[74], https://psl.noaa.gov/data/gridded/data.cobe2.html. SST data were interpolated on a 0.25° latitude × 0.25° longitude grid on a monthly scale from 1850 to 2019.

*Bathymetry*. Bathymetry (m) came from GEBCO Bathymetric Compilation Group 2019 (The GEBCO_2019 Grid—a continuous terrain model of the global oceans and land). Data are provided by the British Oceanographic Data Centre, National Oceanography Centre, NERC, UK. doi:10/c33m. (https://www.bodc.ac.uk/data/published_data_library/catalogue/10.5285/836f016a-33be-6ddc-e053-6c86abc0788e/). These data were interpolated on a 0.25° latitude × 0.25° longitude grid.

*Biological data*. Daily mass concentration of chlorophyll-a in seawater (mg/m³) originated from the Glob Colour project (http://www.globcolour.info/). The product merges together all the daily data from satellites (MODIS, SeaWIFS, VIIRS) available from September 1997 to December 2019, on a 4 km resolution spatial grid. These data were interpolated on a daily scale on a 0.25° latitude × 0.25° longitude grid. These data were only used to map the average maximum standardised SSB (mdSSB) around the North Sea (Fig. 1a). When long-term changes in mdSSB were examined, we used modelled chlorophyll data (see section "Climate projections" below).

Cod recrutement at age 1, Spawning Stock Biomass (SSB) and fishing effort F for 1963–2019 originated from ICES[35].

We used a plankton index of larval cod survival, which was an update of the index proposed by Beaugrand and colleagues[33]. Based on data from the Continuous Plankton Recorder (CPR)[75], the index is based on the simultaneous consideration of six key biological parameters important for the diet and growth of cod larvae and juveniles in the North Sea:[76,77] (i) Total calanoid copepod biomass as a quantitative indicator of food for larval cod, (ii) mean size of calanoid copepods as a qualitative indicator of food, (iii-iv) the abundance of the two dominant congeneric species *Calanus finmarchicus* and *C. helgolandicus*, (v) the genus *Pseudocalanus* and (vi) the taxonomic group euphausiids. A standardised Principal Component Analysis (PCA) is performed on the six plankton indicators for each month from March to September for the period 1958–2017 (table 60 years × 7 months-6 indicators). The plankton index is simply the first principal component of the PCA[33].

*Climate projections*. Climate projections for SST and mass concentration of chlorophyll in seawater (kg m⁻³) originated from the Coupled Model Intercomparison Project Phase 6 (CMIP6)[5] and were provided by the Earth System Grid Federation (ESGF). We used the projections known as Shared Socioeconomic Pathways (SSP) 245 and 585 corresponding respectively to a medium and a high radiative forcing by 2100 (2.5 W m⁻² and 8.5 W m⁻²)[78]. The daily simulations of four different models (i.e. CNRM-ESM2-1, GFDL-ESM4, IPSL-CM6A-LR, and UKESM1-0-LL) covering the time period 1850–2014 (historical simulation) and 2015–2100 (future projections for the two SSPs scenarios) were used. All the data were interpolated on a 0.25° by 0.25° regular grid. Key references (i.e. DOI and dataset version) are provided in Supplementary Text 1. Long-term changes in modelled SSB were based on these data (including modelled daily chlorophyll data).

**The FishClim model**. Let $K_t$ be the maximum standardised Spawning Stock Biomass (mdSSB hereafter) that can be reached by a fish stock at time $t$ for a given environmental regime $\varphi_t$. $X_{t+1}$, standardised SSB (dSSB hereafter) at time $t+1$ was calculated from dSSB at time $t$ as follows:

$$X_{t+1} = X_t + rX_t\left(1 - \frac{X_t}{K_t}\right) - \alpha X_t \tag{1}$$

$\alpha$ is the fishing intensity that varies between 0 (i.e. no fishing) and 1 (i.e. 100% of SSB fished in a year). It is important to note that $\alpha$ (see Eq. (10)) should not be

mistaken with ICES fishing effort F[79] (calculated from SSB). The second term of Eq. (1) is the intrinsic growth rate of the fish stock that is a function of both $K_t$ and the population growth rate r (r was fixed to 0.5 in most analyses, but see Fig. 3d however where r varied from 0.25 to 0.75). The population growth rate r is highly influenced by the life history traits of a species[80] but also by environmental variability[54,55,81]. Here, the population growth rate was assumed to be constant in space and time and the influence of environmental variability occurred exclusively through its effects on $K_t$. We made this choice to not multiply the sources of complexity and errors (i.e. population growth rate is very difficult to assess and varies with age[80]). The third term reflects the part of dSSB that is lost by fishing. Note that natural mortality is not explicitly integrated in Eq. (1) because this process is difficult to assess with confidence at the scale of our study. Here, we assumed that the second term of Eq. (1) implicitly considered this process; when K increases, it is likely that natural mortality diminishes, especially at age 1[34]. We tested this assumption below. Most of the time when fishing occurs, $X_t<K$. But in case of a strong negative environmental forcing at a time of small fishing intensity, $X_t$ can be transitory above K.

Maximum dSSB $K_t$ at time $t$ was assessed using a niche model based on the MacroEcological Theory on the Arrangement of Life (METAL)[82] using SST, an index of food availability based on daily mass concentration of chlorophyll in seawater and bathymetry. The model was therefore based on a three-dimensional niche: thermal, bathymetric and trophic niches.

The thermal niche was asymmetrical. Asymmetric niches can be modelled by using a Gaussian function[83] with the same ecological optimum $y_{opt}$ but two different standard deviations $t_1$ and $t_2$, i.e. two different ecological amplitudes:

$$U_1(y) = ce^{\frac{-(y-y_{opt})^2}{2t_1^2}} \quad \text{When } y \le y_{opt} \tag{2}$$

$$U_1(y) = ce^{\frac{-(y-y_{opt})^2}{2t_2^2}} \quad \text{When } y > y_{opt} \tag{3}$$

Here $y_{opt} = 5.4\,°C$ and $t_1$ and $t_2$ were fixed to 5.7 °C and 4 °C, respectively, so that the thermal niche was close to that assessed by Beaugrand and colleagues[31] (Supplementary Fig. 2). This Supplementary Figure compares the thermal response curve we chose in the present study with the data analysed in Beaugrand and colleagues[31]. The figure shows that the response curve (red curve) is close to the histogram showing the number of geographical cells with a cod occurrence as a function of temperature varying between −2 °C (frozen seawater) and 20 °C.

Because $t_1 > t_2$, the niche was slightly negative asymmetrical (Supplementary Fig. 1). $U_1(y)$ was the first component of mdSSB along the thermal gradient y. c was the maximum value of mdSSB; c was fixed to 1 so that mdSSB varied between 0 and 1[84,85]. y was the value of SST. Slight variations in the different parameters of the niche did not alter either the spatial patterns in the distribution of mdSSB nor the correlations with recruitment.

To model the bathymetric niche of cod, we used a trapezoidal function. Changes in mdSSB, $U_2$, along bathymetry, were assessed using four points ($\theta_1$, $\theta_2$, $\theta_3$, $\theta_4$):

$$U_2(z) = 0 \quad \text{When } z \le \theta_1 \tag{4}$$

$$U_2(z) = \frac{z-\theta_1}{\theta_2-\theta_1}c \quad \text{When } \theta_1 < z \le \theta_2 \tag{5}$$

$$U_2(z) = c \quad \text{When } \theta_2 < z < \theta_3 \tag{6}$$

$$U_2(z) = \frac{\theta_4-z}{\theta_4-\theta_3}c \quad \text{When } \theta_3 \le z < \theta_4 \tag{7}$$

$$U_2(z) = 0 \quad \text{When } z \ge \theta_4 \tag{8}$$

With $\theta_2 \ge \theta_1$, $\theta_3 \ge \theta_2$ and $\theta_4 \ge \theta_3$ and y the bathymetry; $\theta_1 = 0$, $\theta_2 = 10^{-4}$, $\theta_3 = 200$ and $\theta_4 = 600$ m (Supplementary Fig. 1). These parameters were retrieved from the litterature[86,87]. Here also c, the maximum abundance reached by the target species was fixed to 1 and $U_2$ varied between 0 and 1. Trapezoidal niches have been used frequently to model the spatial distribution of fish and marine mammals[88,89].

The trophic niche was modelled by a rectangular function on a daily basis. To the best of our knowledge, no information on the trophic niche is available. We modelled the trophic niche by fixing $U_3$ to 1 when chlorophyll-a concentration was higher than 0.05 mg m⁻³ during a minimum period of 15 days and 0 otherwise (Supplementary Fig. 1). This minimum of chlorophyll was implemented as a proxy for suitable food, which has been shown to be important in the North Atlantic for cod recruitment and distribution[6,33].

There exists two ways to combine the different ecological dimensions of a niche: (i) use an additive or (ii) a multiplicative model[82,90]. We used a multiplicative model because when one dimension is associated to a nil abundance, the resulting abundance combining all dimensions is also nil in contrast to an additive model; therefore only one unsuitable environmental value may explain a nil abundance. All dimensions were associated to abundance values that varied between 0 and 1[90].

Therefore, maximum dSSB, $K$, for a given environmental regime $E$ was given by multiplying the three niches (thermal, bathymetric and trophic):

$$K = \prod_{i=1}^{p} U_i \qquad (9)$$

where $p = 3$, the three dimensions of the niche.

### Analyses

*Mapping of maximum standardised SSB.* mdSSB is close to the "dynamic $B_0$" approach; $B_0$ is the SSB in the absence of fishing (generally expressed in tonnes)[51] whereas mdSSB is the SSB in the absence of fishing standardised between 0 and 1 and assessed from the knowledge of the niche of the species. We first assessed mdSSB in the North-east Atlantic (around UK) at a spatial resolution of 0.25° latitude × 0.25° longitude on a daily basis from 1850 to 2019. For this analysis, FishClim was run on monthly COBE SST (1850–2019), mean bathymetry and a climatology of daily mass concentration of chlorophyll-a in seawater from the Glob Colour project (see Data section). We then calculated an annual average based on the main seasonal productive period around UK, i.e. from March to October[90]. Finally, we averaged all years to examine spatial patterns in mean mdSSB (Fig. 1a).

*Temporal changes in maximum standardised SSB.* We assessed average long-term changes in mdSSB in the North Sea (51°N–62°N and 3°W–9.5°E); the annual average was calculated from March to October because this is a period of high production[90]. We compared long-term changes in mdSSB with cod recruitment at age 1, a plankton index of larval cod survival based on the period March to October[33], and ICES-based SSB[35] for 1963-2019 (Fig. 1b–d).

*Correlation analyses with modelled maximum standardised SSB.* Pearson correlations between long-term changes in mdSSB (average for the North Sea, 51°N–62°N and 3°W–9.5°E) and cod recruitment at age 1 in decimal logarithm[35], a plankton index of larval cod survival in the North Sea[33], and observed ICES SSB in decimal logarithm[35] for the period 1963–2019 were calculated (Fig. 1b–d). The same analysis was performed between assessed fishing intensity α from our FishClim model and fishing effort F[35] in the North Sea (Fig. 1e). The probability of significance of the coefficients of correlation was adjusted to correct for temporal autocorrelation[91].

*Assessment of fishing intensity from ICES spawning stock biomass.* Using North Sea ICES SSB, we applied Eq. (1) to assess fishing intensity α:

$$\alpha = 1 + r\left(1 - \frac{X_t}{K_t}\right) - \frac{X_{t+1}}{X_t} \qquad (10)$$

With $X_{t+1}$ and $X_t$ the ICES dSSB (in decimal logarithm). Standardisation of ICES SSB, necessary for this analysis, was complicated because many different kinds of standardisation were achievable so long as $X$ remained strictly above 0 (i.e. full cod extirpation, not observed so far[35]) and strictly below min(K) (i.e. all black curves always below all points of the blue curve were possible, Supplementary Fig. 3). Indeed, ICES SSB includes exploitation and environmental fluctuations whereas K (i.e. mdSSB) integrates only environmental forcing; the difference is mainly caused by the negative influence of fishing. We chose the black curve (ICES SSB) that maximised the correlation between α (fishing intensity in the FishClim model) and F (ICES fishing effort)[35].

*Reconstruction of long-term changes in ICES spawning stock biomass.* The estimation of α allowed us to reconstruct long-term changes in cod (ICES) dSSB and to examine the respective influence of fishing and CIEC by means of Eq. (1) ("Methods") using four hypothetical scenarios (Fig. 1f). First, we fixed fishing intensity and considered exclusively environmental variations through its influence on dSSB. (i–ii) We assessed long-term changes in dSSB from long-term variation in observed mdSSB (called $K_t$ in Eq. (1)) with a constant level of exploitation fixed to (i) minimum (upper blue curve, i.e. the lowest fishing intensity observed in 1963–2019) or (ii) maximum (lower blue curve, i.e. the highest fishing intensity observed in 1963–2019).

Second, we fixed the environmental influence on dSSB and considered variations in fishing intensity. We estimated long-term changes in dSSB from long-term variation in estimated α with a constant mdSSB fixed to (iii) minimum (lower red curve, i.e. the lowest mdSSB observed in 1963–2019) or (iv) maximum (upper red curve, i.e. the highest mdSSB observed in 1963–2019). It was possible to compare long-term changes in reconstructed (ICES) dSSB (thick black curve in Fig. 1f) with these four hypothetical scenarios (Fig. 1f); note that these comparisons were not affected by the choice we made earlier on the standardisation of (ICES) SSB.

### Quantification of the respective influence of fishing and climate/environment on spawning stock biomass.
Using the previous curves, we examined the respective influence of fishing and CIEC on reconstructed (ICES) dSSB (Fig. 2). First, the influence of fishing was investigated by estimating the residuals between reconstructed (ICES) dSSB based on long-term changes in mdSSB (i.e. $K_t$ in

Eq. (1)) and α (thick black curves in Fig. 1f) and modelled dSSB based on fluctuating fishing intensity α and invariant K (average of the two red curves in Fig. 1f). This calculation led to the red curve in Fig. 2b. Next, we performed the opposite procedure to examine the influence of CIEC on dSSB (i.e. invariant fishing intensity α based on the two blue curves in Fig. 1f). This calculation led to the blue curve in Fig. 2b.

A cluster analysis, based on a matrix years × three time series with (i) long-term changes in reconstructed standardised (ICES) SSBs, (ii) fishing and (iii) CIEC, was performed to identify key periods (vertical dashed lines in Fig. 2). We standardised each variable between 0 and 1 and used an Euclidean distance to assess the year (1963–2019) × year (1963–2019) square matrix so that each variable contributed equally to each association coefficient. We used an agglomerative hierarchical clustering technique using average linkage, which was a good compromise between the two extreme single and complete clustering techniques[92]. In this paper, we were only interested in the timing between the different time periods (i.e. the groups of years) revealed by the cluster analysis (Fig. 2).

We also calculated an index of fishing influence (ε, expressed in percentage) by means of two indicators γ and δ, which were slightly different to the ones we used above. The first one, γ, was modelled dSSB with fluctuating fishing intensity and a constant mdSSB based on the best suitable environment observed during 1963–2019 (only the upper red curve in Fig. 1f; fishing influence). The second one, δ, was modelled dSSB based on fluctuating environment and fishing intensity (black curve in Fig. 1f) on modelled dSSB based on a fluctuating environment but a constant fishing intensity fixed to the lowest value of the time series (only the upper blue curve in Fig. 1f; environmental influence). The index of fishing influence (ε, expressed in percentage) was calculated as follows:

$$\varepsilon = \frac{100\gamma}{\gamma + \delta} \qquad (11)$$

For each period of 1963–2019 identified by the cluster analysis, we quantified the influence of fishing (and therefore the environment) using a Jackknife procedure[93,94]. The resampling procedure recalculated ε by removing each time 1 year of the time period, which allowed us to provide a range of values (i.e. minimum and maximum) in addition to the average value $\bar{\varepsilon}$ calculated for each interval, including the whole period (Fig. 2c).

*Long-term changes in modelled spawning stock biomass (1850–2019, 2020–2100 and 2020-2300).* We modelled mdSSB ($K_t$ in Eq. (1)) using outputs from four Earth System models (ESMs) based on two scenarios of SST/Chlorophyll changes (i.e. SSP245 and SSP585) for the period 1850–2100 (and for one scenario and one ESM until 2300; Fig. 3).

For the period 1850–2019, we used daily SST/Chlorophyll changes from the four ESMs to estimate potential changes in mdSSB (thin dashed black curves in Fig. 3a). An average of mdSSB was also calculated (thick green curve in Fig. 3a).

For the period 2020–2100, we showed all potential changes in mdSSB based on the four ESMs and both scenarios SSP245 (thin dashed blue curves in Fig. 3a) and SSP585 (thin dashed red curves). An average of mdSSB was also calculated for scenarios SSP245 (thick continuous blue curve) and SSP585 (thick continuous red curve). In addition, we assessed dSSB based on a constant standardised catch fixed to the average of 2008–2019, the last period identified by the cluster analysis (G5, i.e. $\alpha X = 0.03$ in Eq. (1)), and the average values of all ESMs for SSP245 (thick dashed blue curve in Fig. 3a) and SSP585 (thick dashed red curve). This analysis was performed to show how a constant catch might alter long-term changes in mdSSB. When $X_t$ (Eq. (1)) reached 0.1, the stock was considered as fully extirpated.

*Understanding how fishing and climate/environment interact now and in the future.* We modelled dSSB as a function of fishing intensity α and CIEC to show how fishing and the environment interact (Fig. 3b, c). We calculated dSSB for fishing intensity between $\alpha = 0$ and $\alpha = 0.5$ every step $\Theta = 0.001$ and for mdSSB between $K = 0$ and $K = 1$ every step $\Theta = 0.001$ to represent values of dSSB as a function of fishing and CIEC. We then superimposed reconstructed ICES dSSB (1963–2019) on the diagram for three periods: 1963–1985 (high SSB), 1986–1999 (pronounced reduction in SSB), and 2000–2019 (low SSB). Maximum standardised SSB for 2020–2100 (or 2300 exclusively for Scenario SSP 585 of IPSL ESM) assessed from four ESMs and scenarios SSP245 and SSP585 were also superimposed. Fishing intensity is unpredictable for 2020–2100 and so we arbitrarily fixed it constant between 0.08 and 0.17 in increments of 0.1 for display purposes, starting by ESMs based on scenario SSP 245 followed by scenario SSP 585 (Fig. 3b). When $X_t$ (Eq. (1)) reached 0.1, the stock was considered as fully extirpated.

We calculated an index of sensitivity of dSSB as a function of fishing intensity and CIEC. To do so, we first calculated sensitivity of dSSB to fishing intensity α. Index $\zeta_i$ was calculated at point $i$ from dSSB $X$ and fishing intensity α at $i-1$ and $i+1$ (see also Eq. (1)):

$$\zeta_i = \frac{|X_{i+1} - X_{i-1}|}{|\alpha_{i+1} - \alpha_{i-1}|} \quad \text{with } \min(\alpha) + \theta \leq i \leq \max(\alpha) - \theta \qquad (12)$$

With $\min(\alpha) = 0$, $\max(\alpha) = 0.5$ and $\Theta = 0.001$.

Similarly, we calculated sensitivity of dSSB to $K$. Index $\eta_j$ was calculated at point $j$ from dSSB $X$ and mdSSB K at $j-1$ and $j+1$ (see also Eq. (1)):

$$\eta_j = \frac{\left| X_{j+1} - X_{j-1} \right|}{\left| K_{j+1} - K_{j-1} \right|} \quad \text{with } \min(K) + \theta \le j \le \max(K) - \theta \tag{13}$$

With $\min(K) = 0$, $\max(K) = 1$ and $\Theta = 0.001$.

Then, we summed the two indices to assess the joint sensitivity of dSSB to fishing intensity $Z$ and mdSSB $H$:

$$\mathbf{I}_{i,j} = \mathbf{Z}(\zeta_i) + \mathbf{H}(\eta_j) \tag{14}$$

Matrix $\mathbf{I}$ was subsequently standardised between 0 and 1:

$$\mathbf{I}^* = \frac{\mathbf{I} - \min(\mathbf{I})}{\max(\mathbf{I}) - \min(\mathbf{I})} \tag{15}$$

With $\mathbf{I}^*$ the matrix of sensitivity of dSSB to fishing intensity and mdSSB standardised between 0 and 1 (Fig. 3c).

*Number of years needed for recovery after stock collapse.* We investigated how the number of years needed for a stock to recover after stock collapse (i.e. dSSB=0.05 in Eq. (1); i.e. 10% of mdSSB) varied as a function of mdSSB (between 0 and 1 by increment of 0.001); this was only influenced by the environmental regime $\varphi_t$ and population growth rate $r$. For this analysis, we fixed a target dSSB of 0.4 (vertical dashed green vertical line in Fig. 3d) and three different values of $r$: 0.25, 0.5 and 0.75. We simulated a hypothetical moratorium with a fishing intensity $\alpha = 0$ in Eq. (1).

Here, stock collapse was defined as dSSB $\le 0.1 \times$ mdSSB, i.e. when the dSSB reached less than 10% of the unfished biomass mdSSB. This threshold corresponds to values usually defined in the literature; e.g. Pinsky and colleagues[95] defined a collapse when landings are below 10% the average of the five highest landings recorded for more than 2 years, Worm and colleagues[69] defined stock collapse when the biomass becomes lower than 10% of the unfished biomass, Andersen[96] when it is lower than 20% and Thorpe and De Oliveira[67] when it is lower than 10–20%.

*Potential consequences of fisheries management and climate-induced environmental changes.* We examined how fishing and CIEC may affect cod stocks and their exploitation around UK with a focus in the North Sea (Figs. 4, 5). For these analyses, we averaged long-term changes in modelled dSSB corresponding to each scenario (all thin dashed blue and thin red curves in Fig. 3a for SSP245 and 585, respectively). In these analyses, the stock was considered fully extirpated when $X_t$ (Eq. (1)) reached 0.1.

Year of cod extirpation for 2020–2100: We estimated year of cod extirpation from 2020 to 2100 in each geographical cell based on (i) a constant fishing intensity ($\alpha = 0.04$) in time and space, and (ii) an adjusted fishing intensity using the concept of Mean Sustainable Yield (MSY). The choice of $\alpha = 0.04$ did not alter our conclusions; a lower or a higher value delayed or speed cod extirpation in a predictable way, respectively.

In fisheries, MSY is defined as the maximum catch (abundance or biomass) that can be removed from a population over an indefinite period with $dX/dt = 0$, with $X$ for dSSB and $t$ for time. Despite some criticisms about MSY[66], the concept remains a key paradigm in fisheries management[35,63]. We used this concept to show that controlling fishing intensity delayed cod extirpation. From Eq. (1), we calculated fishing intensity, called $\alpha_{MSYt}$, so that $X$ remained above $X_{MSYt}$ at all time $t$:

$$\alpha_{MSYt} = r \left( 1 - \frac{X_{MSYt}}{K_t} \right) \tag{16}$$

In this analysis, we fixed $X_{MSY\,t} = K_t/2$.

We assessed $\alpha_{MSYt}$ from Eq. (16) and then estimated dSSB from $\alpha_{MSYt}$ and $K_t$ (based on averaged SSP245 and SSP585) by means of Eq. (1).

Although results were displayed at the scale of the north-east Atlantic (around UK), we calculated the difference in year of cod extirpation between scenarios of warming (SSP245 and SSP585) and between scenarios of cod management (constant versus adjusted—MSY— fishing intensity). Differences were presented by means of histograms (Fig. 4). From each histogram, we calculated the median of the differences in year of cod extirpation $E$[97].

Pooled standardised catch by 2100 (2020–2100): In term of fishing exploitation, we assessed pooled standardised catch (i.e. pooled dSSB) in 2100 (2020–2100), again for two scenarios of CIEC (SSP245 and 585) and two scenarios of cod management (constant versus adjusted—MSY—fishing intensity; Fig. 5). We then calculated the percentage of reduction in pooled standardised catch caused by fishing or the intensity of warming. Finally, we assessed the median of the percentage of reduction in pooled standardised catch for the North Sea area (51°N–62°N and 3°W–9.5°E). The goal of this analysis was to demonstrate that controlling fishing intensity optimises cod exploitation.

**Statistics and reproducibility.** All statistical analyses can be reproduced from the equations provided in the text, the cited references or the data available in Supplementary Data.

**Reporting summary.** Further information on research design is available in the Nature Research Reporting Summary linked to this article.

## Data availability

The main data used in this paper are in Supplementary Data and other data are available from the corresponding author on reasonable request.

## Code availability

All codes used in this paper are available from the corresponding author on reasonable request.

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

## Acknowledgements

The authors acknowledge the World Climate Research Programme, which, through its Working Group on Coupled Modelling, coordinated and promoted CMIP6. We thank the climate modelling groups for producing and making available their model output, the Earth System Grid Federation (ESGF) for archiving the data and providing access, and the multiple funding agencies who support CMIP6 and ESGF. This work has been partially financially supported by Université du Littoral Côte d'Opale, France as part of the IFSEA Graduate School, and by the CPER IDEAL.

## Author contributions

G.B. conceived the study. G.B. and A.B. designed the models. G.B., A.B. and L.K. prepared the data and performed the analyses. G.B. prepared the first draft. G.B., R.R.K., A.B. and L.K. discussed the results and contributed to the writing.

## Competing interests

The authors declare no competing interests.
