## [Peer Review File · Communications Biology]

REVIEWERS' COMMENTS:

Reviewer #1 (Remarks to the Author):

Dear Dr Beaugrand and co-authors,

I am pleased to see that you have done a thorough and good job in enhancing your manuscript based upon input from me and the other reviewers. You have also argued well for your views where you have not followed up on my input. In most cases you have convinced me that you are right, or at least not wrong (there may be different views).

I still do not at all like where your wording suggests that there is an ongoing discussion of whether fisheries or climate/environment drive the fluctuations in fish stocks. Neither of the two papers you point to claim that only one of the factors is important, only that for some stocks and some periods the one or other dominates (ask Keith or Max about it!). However, your claim that there still is such a dichotomy is now strongly phrased only in your rebuttal letter (which I'm not reviewing). I find the wording in your paper has been modified and it is now acceptable to me. Congratulations with a good and interesting paper!

Reviewer #1 (Remarks to the Author):

Reviewer 1 said “In this study, the authors use the “FishClim” model to consider the relative contributions of fishing and climate-induced change on the North Sea cod stock, finding that both have been important since 1963, and they argue that we need to move beyond paradigms that consider either fishing or climate in isolation to consider them both together. I agree with the need for the work, and the general conclusion, and the model itself contains interesting and novel elements.”

Thank you.

Reviewer 1 said “However, I found that the manuscript was uneven and frustrating in parts, and needs significant improvement before publication could be recommended. In my view the key problems are:- a) Opacity of method at various parts of the manuscript that makes it hard to follow exactly what has been done.”

We have clarified our methods and think the revised ms is clearer (see our answers below). We thank the reviewer for her/his comments.

Reviewer 1 said “b) Assertions that need more evidential support and cannot just be taken at face value.”

This point has been carefully considered in the revision. We respond in details in the specific answers below and have in all cases considered the comments made by the reviewer.

Reviewer 1 said “c) Insufficient discussion of the weaknesses of the modelling approach.”

We have a new section that discusses potential caveats of our approach. **We say line 296 page 9:**

“The FishClim model is structured in space and time, and includes both fishing and environmental effects, which make it possible to assess the respective influence of the two drivers in space and time. Although our model assesses a standardised SSB that could subsequently be scaled to the actual SSB of a stock, it does not include information on its size/age structure, which is considered to be important for management purposes¹. In addition, our present version of the model does not include natural mortality because this process is difficult to assess with confidence at the scale of our study²; here we assumed it is integrated into the second term of Equation 1 (Methods).

The two time-series of fishing intensity (α in our model) and effort (ICES F) were significantly correlated positively (Fig. 1e). They exhibited similar long-term patterns with a pronounced increase in fishing intensity α and effort F in the mid-1960s, a strong reduction in the mid-2000s and high values between these two periods. In addition, low periods of fishing intensity and effort were observed at the beginning and the end of the time period (before the mid-1960s and after the mid-2000s). The medium correlation, although significant, was mainly due to year-to-year variance in the estimations of fishing intensity/effort that might originate from the difficulty in estimating such parameters³. Nevertheless, given the different methods used to assess fishing intensity (α) and effort (ICES F), we think that it is reassuring that the two time-series exhibit similar long-term changes (Fig. 1e).

Although we assessed the influence of r on timing for recovery after a hypothetical collapse (see Fig. 3d), we performed most analyses with a constant population growth rate $r=0.5$. Population growth

rate r is likely to be affected by temperature and food availability^{4, 5} but it remains strongly determined by the life history traits of a species. For example, r would be higher for a r - than a K -strategy species⁶. Nevertheless, a dynamic r might be easily employed in our model but it is difficult to know how temperature and chlorophyll concentration may jointly affect r and in practice, it might be difficult to implement realistic changes in r ⁴.

Migration was not accounted for into the model. In this study, we assumed that (i) migration had a small influence on standardised SSB at the scale of the North Sea⁷. Although some studies have suggested that cod migration was limited to 500 km at maximum⁸, more recent estimates suggest that this value was perhaps too extreme⁹. More recent studies found that in the summer (mid-June to mid-August) the range of cod movement was less than one km¹⁰. Evidence from electronic tagging experiments also suggests that there were behavioural differences between the English Channel and the North Sea cod that limit their mixing in the two areas⁹. Other mark and recapture experiments, as well as genetic evidence, have suggested that populations from the northern North Sea (>57°N) did not intermix significantly with those from the southern North Sea (<56°N)^{11, 12}.

Although being largely debated for decades because of uncertainties on the estimates (e.g. lack of reliability and poor assumptions in some models) or because it is too specific and does not include other fisheries^{13, 14, 15, 16, 17, 18}, we chose to use B_{MSY} because it remains widely used by agencies regulating fisheries and in North Sea cod management^{17, 19}. However, our model can be employed with any biological reference points such as those currently discussed in the literature^{18, 20}.

Multispecies Maximum Sustainable Yield (MMSY) is being increasingly used²¹. The effect of multispecies fishing in our model would be to lead to an underestimate of α . This potential issue could be partially solved by subdividing α into two components α_1 (i.e. direct fishing effect) and α_2 (indirect fishing effect). MMSY remains not easy to implement at the organisational community level because it is challenging to maximise all stocks simultaneously and inevitably there are some stocks that might be overfished while others might be underfished¹⁷. Nevertheless, our approach based on B_{MSY} remains important because our model proposes a dynamic MSY that is adjusted as a function of environmental changes.

To estimate the maximum standardised Spawning Stock Biomass ($mdSSB$, K_t in Equation 1), our FishClim model used an empirical niche model, i.e. a multiplicative empirical model that integrates temperature, bathymetry and chlorophyll concentration (duration and concentration). Although the niche is composed of more ecological dimensions, the three chosen parameters are key for fish distribution^{22, 23}. The values of the different parameters of the niche were fixed according to our knowledge of the fish^{24, 25, 26} and slight modifications in the values of these parameters did not alter our conclusions. Our models could be forced by any ecological niche models (or species distribution models) such as the Non-Parametric Probabilistic Ecological Niche Model (NPPEN) or the Maximum Entropy (MaxEnt) model to assess $mdSSB$ ^{24, 27}.

Inter-ESM variability remains important and it is clear that this affects our projections (Fig. 3-5). In addition, emission scenarios are inherently unpredictable and this might also influence our projections, although in more expected ways (Fig. 4-5). However, the model we propose could be used on a year-to-year basis to better anticipate future changes in SSB and predict more realistic fishing quotas that may either prevent stock collapse or better optimise exploitation. “

Reviewer 1 said “d) Insufficient framing of the study in the context of other work. There is other work on modelling future impacts of climate and fishing together in the North Sea, and on determination of dynamic B0 (analogous to $mdSSB$) – some references added.”

We have added and discussed all references, and some more, provided by the reviewer. We thank the reviewer for the references.

Reviewer 1 said “I hope that the authors will be able to improve the paper by addressing these shortcomings.”

We have carefully considered all comments made by the reviewer and think the paper has greatly improved as a result. We thank the reviewer.

Reviewer 1 said “L23 : “can immediately adjust fishing effort in response to environmentally-driven shifts in stock productivity”..

Modified directly in the revision.

Reviewer 1 said “L25-26 : 3 significant figures is too much precision here, ~55/45 split might be better.”

Modified in the revision.

Reviewer 1 said “L26 – I agree with this conclusion but wish the study could support it more comprehensively.”

We have clarified the methods in the revision. See our above responses.

Reviewer 1 said “L33 – process not processes.”

Modified in the revision.

Reviewer 1 said “L40 I found references to dSSB and mdSSB confusing as surplus production models normally talk about carrying capacity (K) and fisheries scientists use B₀ (virgin biomass) or dynamic B₀ (Beasell -Browne et al. 2022; Punt et al., Amar et al., 2009, Punt et al., 2014) (environment-mediated virgin biomass). It would be better to use these terms in the paper.”

We think our acronym mdSSB should be distinguished from B₀ because it is not a biomass in the absence of fishing but rather a standardised biomass (varies between 0 and 1) in the absence of fishing. It is assessed from the knowledge of the ecological niche of cod. Therefore, the two indices are different. We have clarified this point in the revision. We say line 902 page 25:

“Maximum standardised SSB, called mdSSB hereafter, is close to the “dynamic B₀” approach; B₀ is the SSB in the absence of fishing (generally expressed in tonnes)²⁰ whereas mdSSB is the SSB in the absence of fishing standardised between 0 and 1 and assessed from the knowledge of the niche of the species. We first assessed mdSSB in the Northeast Atlantic (around UK) at a spatial resolution of 0.25° latitude x 0.25° longitude on a daily basis from 1850 to 2019. For this analysis, FishClim was run on monthly COBE SST (1850-2019), mean bathymetry and a climatology of daily mass concentration of chlorophyll-a in sea water from the Glob Colour project (see Data section)...”

Reviewer 1 said “L42 – effectively this is a grid of surplus production models. The strengths and weaknesses of this choice need to be discussed somewhere with much greater rigour (see attached annex A). The approach is novel to me, but there are some problems, not least that all the climate impact comes from changing K, whilst r is not a function of climate. That is hard to justify (brander, 1995; Rountrey et al., 2014)”

We now discuss in the revision the potential caveats of our approach line 296 page 9. See the new paragraph above in our answers to the reviewer.

We have also added a few lines in the Methods section. **We say line 839 page 23:**

“The population growth rate r is highly influenced by the life history traits of a species²⁸ but also by environmental variability^{4, 5, 29}. Here, the population growth rate was assumed to be constant in space and time and the influence of environmental variability occurred exclusively through its effects on K_t . We made this choice to not multiply the sources of complexity and errors (i.e. population growth rate is very difficult to assess and varies with age²⁸).”

Reviewer 1 said “L51 – how is the annual average determined from the March-October period alone? More details on this are needed, perhaps in supplementary.”

This is also because the plankton index of larval cod survival is based on the same period for the same reason. We have clarified this point in the main text **line 912 page 25:**

“...the annual average was calculated from March to October because this is a period of high production³⁰. We compared long-term changes in $mdSSB$ with cod recruitment at age 1, a plankton index of larval cod survival³¹, and observed ICES SSB¹⁹ for 1963-2019 (Fig. 1b-d).”

Reviewer 1 said “L53 delete “expectedly”

Deleted from the revision.

Reviewer 1 said “L53-55 – what about for lag 0 and 2? Is a lag of 1 biologically justfield?”

This point has been carefully investigated in Beaugrand and Kirby²⁶ by means of an crosscorrelogram (their Figure 1f). There was a significant lag of one year between recruitment at year 1 and plankton changes in the North Sea. This correlation was statistically significant and biologically expected. We have clarified this point in the revision **line 78 page 4.**

“In addition to be expected biologically because recruitment is assessed at age 1, the one-year lag was also found in some studies that investigated relationships between changes in plankton and cod recruitment^{26, 31}.”

Reviewer 1 said “L59 not sure about this statement. Natural mortality tends to decrease with size (Andersen, 2020) so should be greatest at year 0. See also Pope et al., 2020”

The reviewer was correct. The symbol “ \leq ” was missing. We have made the correction in the revision.

Reviewer 1 said “L66 – see methods – this section is totally opaque and is in need of a complete re-write to provide details about the nature of the standardisation used.”

We have clarified this point in the Methods section **lines 928-936 page 25:**

“Standardisation of ICES SSB, necessary for this analysis, was complicated because many different kinds of standardisation were achievable so long as X remained strictly above 0 (i.e. full cod extirpation, not observed so far¹⁹) and strictly below $\min(K)$ (i.e. all black curves always below all points of the blue curve were possible, Supplementary Fig. 2). Indeed, ICES SSB includes exploitation and environmental fluctuations whereas K (i.e. $mdSSB$) integrates only environmental forcing; the difference is mainly caused by the negative influence of fishing. We chose the black curve (ICES SSB) that maximised the correlation between α (fishing intensity in the FishClim model) and F (ICES fishing effort)¹⁹.”

Reviewer 1 said “L66-69. ICES SSB contains implicit information about ICES F, so it is not surprising that fishing intensity and ICES F are related. The degree of relation ought to be a function of the impact of environmental shifts in this model relative to those that influenced SSB. I think I might have expected a better correlation than 0.56.”

We do not think that we should have expected a correlation higher than 0.56 because the two methodologies are different. Nevertheless, we now discuss this point in the discussion section lines 304-313 page 10:

“The two time-series of fishing intensity (α in our model) and effort (ICES F) were significantly correlated positively (Fig. 1e). They exhibited similar long-term patterns with a pronounced increase in fishing intensity α and effort F in the mid-1960s, a strong reduction in the mid-2000s and high values between these two periods. In addition, low periods of fishing intensity and effort were observed at the beginning and the end of the time period (before the mid-1960s and after the mid-2000s). The medium correlation, although significant, was mainly due to year-to-year variance in the estimations of fishing intensity/effort that might originate from the difficulty in estimating such parameters³. Nevertheless, given the different methods used to assess fishing intensity (α) and effort (ICES F), we think that it is reassuring that the two time-series exhibit similar long-term changes (Fig. 1e)”.

Reviewer 1 said “L90-100 again this is hard to follow. There is not enough information about how the periods were segmented, though I do agree it shows both climate and fishing are important. See also Caswell et al., 2020 North Sea case study, which seems to suggest cod ought to have recovered given reductions in F and assuming a constant environment – the implication is that degradation of the environment for cod has prevented this.”

We have clarified this point in the revision in the Methods section. We say lines 262-970 page 26:

“A cluster analysis, based on a matrix years \times three time series with (i) long-term changes in reconstructed standardised (ICES) SSBs, (ii) fishing and (iii) CIEC, was performed to identify key periods (vertical dashed lines in Fig. 2). We standardised each variable between 0 and 1 and used an Euclidean distance to assess the year (1963-2019) \times year (1963-2019) square matrix so that each variable contributed equally to each association coefficient. We used an agglomerative hierarchical clustering technique using average linkage, which was a good compromise between the two extreme single and complete clustering techniques³². In this paper we were only interested in the timing between the different time periods (i.e. the groups of years) revealed by the cluster analysis (Fig. 2).”

We also cite two news references that was mentioned in the review of Caswell: MacKenzie et al 2011³³ and Eero et al 2011³⁴.

Reviewer 1 said “L110-126 – stick with 2 sig fig for all the %ages.”

This has been done directly in the revision.

Reviewer 1 said “L145-146 – how is stock collapse defined, and how does it compare with the literature?”

We have homogenised the threshold for collapse so that it is now in agreement with the values used in the literature. We have redone figure 3d accordingly. We now say in the revision lines 1037-1042-716 page 28:

“Here, stock collapse was defined as $dSSB \leq 0.1 \times mdSSB$, i.e. when the $dSSB$ reached less than 10% of the unfished biomass $mdSSB$. This threshold corresponds to values usually defined in the literature; e.g. Pinsky and colleagues³⁵ defined a collapse when landings are below 10% the average of the five highest landings recorded for more than two years, Worm and colleagues²¹ defined stock collapse when the biomass becomes lower than 10% of the unfished biomass, Andersen³⁶ when it is lower than 20% and Thorpe & De Oliveira¹⁷ when it is lower than 10-20%.”

Reviewer 1 said “L149-169 there is a need somewhere to compare this with other studies that consider warming and fishing.”

We now discuss these results in a paragraph that has been added in the discussion section **lines 259-274 page 9**. We say:

“To our knowledge, a few studies have examined the joint influence of climate change and fishing on cod^{34, 37, 38}. Engelhard and colleagues³⁷ have investigated the influence of both drivers on the spatial distribution of cod in the North Sea over the past 100 years. The authors showed that the deepening and northward shift of cod was attributable to warming whereas the eastward shift was best explained by fishing that strongly depleted the stock off the coasts of England and Scotland. Their study revealed the fundamental importance of both climate change and fishing pressure for our understanding of North Sea cod. In the Baltic Sea, a study investigated changes through time of the respective influence of CIEC, predation, eutrophication and exploitation on cod biomass during the 20th century and quantified their respective influence³⁴. At the beginning of the 20th century, nutrient availability and mammal predation were the main drivers of the size of the stock. Then from the 1940s, fishing became dominant. In the 1980s, eutrophication starts to play a role. For the period 1980-1984, the authors assessed that the relative influence of eutrophication, climate and fishing on Baltic cod biomass was 13%, 43% and 52%, respectively³⁴. Although we did not assess the potential influence of eutrophication on North Sea cod because this sea is only marginally influenced by this environmental issue³⁹, our estimates for fishing and climate were >70% and <30% for fishing and CIEC in the North Sea, respectively (Fig. 2c).”

Reviewer 1 said “L171 – “collapse” is not normally taken to be 5% of B_0 , (e.g. 10% in Worm et al. 2009; see also discussion in Thorpe and De Oliviera, 2019).”

We have homogenised our thresholds throughout the paper. Note this does not affect our conclusions. As a result a new figure 3d has been performed but the figure is very close to the previous one based on 0.05. As said by the reviewer, 10% is more widely used in the literature. We have also better justified this choice by using the references provided by the reviewer (see our answer below).

Reviewer 1 said “L224-253. I agree with the conclusions, however to make them stronger, they should be discussed in the light of other studies of fishing and climate, and a discussion of management to dynamic B_0 must be part of this.”

We agree with the reviewer. We have a paragraph in the discussion **lines 259-274 page 9**. See the above paragraph.

We have included the dynamic B_0 concept in the Methods section **lines 922-925 page 25**:

“Maximum standardised SSB, called $mdSSB$ hereafter, is close to the “dynamic B_0 ” approach; B_0 is the SSB in the absence of fishing (generally expressed in tonnes)²⁰ whereas $mdSSB$ is the SSB in the

absence of fishing standardised between 0 and 1 and assessed from the knowledge of the niche of the species.”

Reviewer 1 said “L473 – switching between K and mdSSB is confusing. Please use K as this is standard in fisheries.”

We separate K, which is a mathematical symbol included in Equation 1, from its meaning that is mdSSB (maximum standardised SSB).

Reviewer 1 said “L479 – fixing r is a major weakness of the study as climate change will change cod metabolism, size structure and growth response. Also a justification for the choice of 0.5 for cod is required. R and K are strongly related in surplus production models, so this choice has implications for the Ks that are allowable.”

To explain why we made this choice, we have also added a few lines in the Methods section. We say lines 860-865 page 23:

“The population growth rate r is highly influenced by the life history traits of a species²⁸ but also by environmental variability^{4, 5, 29}. Here, the population growth rate was assumed to be constant in space and time and the influence of environmental variability occurred exclusively through its effects on K_t . We made this choice to not multiply the sources of complexity and errors (i.e. population growth rate is very difficult to assess and varies with age²⁸).”

Reviewer 1 said “L486-520 – I thought the discussion of niches was really interesting.”

Thank you.

Reviewer 1 said “L495 – can you explain why these values were chosen? How do they compare with BIOMOD or MAXENT niches?”

We have clarified these points in the revision in the discussion section. We now say lines 352-360 page 11:

“To estimate the maximum standardised Spawning Stock Biomass (mdSSB, K_t in Equation 1), our FishClim model used an empirical niche model, i.e. a multiplicative empirical model that integrates temperature, bathymetry and chlorophyll concentration (duration and concentration). Although the niche is composed of more ecological dimensions, the three chosen parameters are key for fish distribution^{22, 23}. The values of the different parameters of the niche were fixed according to our knowledge of the fish^{24, 25, 26} and slight modifications in the values of these parameters did not alter our conclusions. Our models could be forced by any ecological niche models (or species distribution models) such as the Non-Parametric Probabilistic Ecological Niche Model (NPPEN) or the Maximum Entropy (MaxEnt) model to assess mdSSB^{24, 27}.”

Reviewer 1 said “L503 – “significant points”?”

We have removed “remarkable” from the revision. We prefer not using “significant” as advised by the reviewer.

Reviewer 1 said “L513 – can you provide any other references in support of this?”

We have added a second reference from the same group.

“Kaschner K, Watson R, Trites AW, Pauly D (2006) Mapping world-wide distributions of marine mammal species using a relative environmental suitability (RES) model. Marine Ecology Progress Series 316:285-310.”

Reviewer 1 said “L520 – this is a really interesting idea. Can you provide 1-2 sentence justification of multiplying the niches together? Does it assume T, trophic, and bathymetry are all equally limiting for cod?”

We have clarified this point in the revision lines 910-915 page 24:

“There exists two ways to combine the different ecological dimensions of the niche: (i) use an additive or (ii) a multiplicative model⁴⁰. We have used a multiplicative model because if one dimension is associated to a nil abundance, the resulting abundance combining all dimensions will be nil in contrast to an additive model; therefore only one unsuitable environmental value may explain a nil abundance. All dimensions were associated to abundance values that vary between 0 and 1.”

Reviewer 1 said “L540-544 it is impossible to follow this section. Why was standardisation needed. What ones were tried, and why was this one chosen? Can you provide more details?”

We have clarified this point in the revision lines 948-954 page xxx. We say:

“Standardisation of ICES SSB, necessary for this analysis, was complicated because many different kinds of standardisation were achievable so long as X remained strictly above 0 (i.e. full cod extirpation, not observed so far¹⁹) and strictly below $\min(K)$ (i.e. all black curves always below all points of the blue curve were possible, Supplementary Fig. 2). Indeed, ICES SSB includes exploitation and environmental fluctuations whereas K (i.e. $mdSSB$) integrates only environmental forcing; the difference is mainly caused by the negative influence of fishing. We chose the black curve (ICES SSB) that maximised the correlation between α (fishing intensity in the FishClim model) and F (ICES fishing effort)¹⁹.”

Reviewer 1 said “L546 – this is quite confusing. Normally harvest rates can be directly calculated from F_s . Is this because the ICES SSB and the nominal SSB from the surplus model are really quite different?”

Our model uses fishing intensity α , which is different to fishing effort because SSB is standardised between 0 and 1; this is imposed by the niche model. We have clarified this point in the revision lines 855-858 page 23:

“ α is the fishing intensity that varies between 0 (i.e. no fishing) and 1 (i.e. 100% of SSB fished in a year). It is important to note that α (see Equation 10 below) should not be mistaken with ICES fishing effort F ⁴¹ (calculated from SSB).”

Reviewer 1 said “L568-575 I am afraid this section is totally incomprehensible to me.”

We have clarified the paragraph in the discussion section lines 982-986 page 26 (see our changes in red).

Reviewer 1 said “L636-641. $dSSB$ would not normally be 0.05 for collapse, see Worm et al., 2009, Thorpe and De Oliveira, 2019, Hilborn 2021. Also what is the biological significance of the 3 different r values.”

$dSSB$ is now fixed to 0.1 throughout the paper, which is more in agreement with the literature. We have just redone figure 3d, which provides similar conclusions.

We used three different r values to see how our predictions were sensitive to uncertainties in r . We have clarified this point in the revision lines 323-329 page 10. We now say:

“Although we assessed the influence of r on timing for recovery after a hypothetical collapse (see Fig. 3d), we performed most analyses with a constant population growth rate $r=0.5$. Population growth rate r is likely to be affected by temperature and food availability^{4,5} but it remains strongly determined by the life history traits of a species. For example, r would be higher for a r - than a K -strategy species⁶. Nevertheless, a dynamic r might be easily employed in our model but it is difficult to know how temperature and chlorophyll concentration may jointly affect r and in practice, it might be difficult to implement realistic changes in r^A .”

”

Reviewer 1 said “L647 – same issue. Why is full extirpation at 0.1 whilst collapse is at 0.05 (L637)? This seems both inconsistent with fisheries practice and internally inconsistent.”

We have redone our analysis for Fig. 3d using a threshold of 0.1 and our interpretation for Fig. 3a-c uses a threshold of 0.1. Note this does not alter our conclusions. See the new Figure 3d. Therefore our threshold of cod collapse is 0.1 for all analyses and interpretations in the revision.

Reviewer 1 said “L650 – again what is the relevance of this scenario? How does such a harvest rate relate to a) the past, or b) MSY management?”

Reviewer 1 said “L655 – add reference to Mesnel (2012) who has a useful history of MSY.”

We agree. The reference has been added in the revision line 341.

Reviewer 1 said “L658 – hard to justify that MSY is at $B_0/2$ – see Punt et al or Thorpe et al., 2015 (from discussion on Wikipedia).”

We have now a paragraph on this point in the discussion section. We say lines 339-351 page 11:

“Although being largely debated for decades because of uncertainties on the estimates (e.g. lack of reliability and poor assumptions in some models) or because it is too specific and does not include other fisheries^{13, 14, 15, 16, 17, 18}, we chose to use B_{MSY} because it remains widely used by agencies regulating fisheries and in North Sea cod management^{17, 19}. However, our model can be employed with any biological reference points such as those currently discussed in the literature^{18, 20}.

Multispecies Maximum Sustainable Yield (MMSY) is being increasingly used²¹. The effect of multispecies fishing in our model would be to lead to an underestimate of α . This potential issue could be partially solved by subdividing α into two components α_1 (i.e. direct fishing effect) and α_2 (indirect fishing effect). MMSY remains not easy to implement at the organisational community level because it is challenging to maximise all stocks simultaneously and inevitably there are some stocks that might be overfished while others might be underfished¹⁷. Nevertheless, our approach based on B_{MSY} remains important because our model proposes a dynamic MSY that is adjusted as a function of environmental changes.

Reviewer 1 said “Figure 1a – add units in words – fraction of local carrying capacity.”

This has been modified in the revised figure.

Reviewer 1 said “Figure 1e – why is fishing intensity so different from ICES F? this suggests a problem with the analysis and must be justified in detail.”

The following discussion has been added in the revision **lines 313-322 page 10:**

“The two time-series of fishing intensity (α in our model) and effort (ICES F) were significantly correlated positively (Fig. 1e). They exhibited similar long-term patterns with a pronounced increase in fishing intensity α and effort F in the mid-1960s, a strong reduction in the mid-2000s and high values between these two periods. In addition, low periods of fishing intensity and effort were observed at the beginning and the end of the time period (before the mid-1960s and after the mid-2000s). The medium correlation, although significant, was mainly due to year-to-year variance in the estimations of fishing intensity/effort that might originate from the difficulty in estimating such parameters³. Nevertheless, given the different methods used to assess fishing intensity (α) and effort (ICES F), we think that it is reassuring that the two time-series exhibit similar long-term changes (Fig. 1e).”

Reviewer 1 said “S. Table 1 – nice table, typo in line 5. MSY – maximum sustainable yield !!”

Modified directly in the revision.

Reviewer 1 said “Could add Carbon brief ref for the SSPs, - see Carbon Brief 2018 – <https://www.carbonbrief.org>”

“Carbonbrief” is not the primary reference for SSPs. We think the reference to Eyring and colleagues (“Eyring V, et al. Overview of the Coupled Model Intercomparison Project Phase 6 (CMIP6) experimental design and organization. *Geoscientific Model Development* 9, 1937-1958 (2016). ») is much more relevant and that there is no need to refer to this website.

Reviewer 1 said “Figure 2 – cf Caswell et al., 2020, North Sea case study, where cod would be expected to recover given a constant environment, but hasn’t.”

Reviewer 1 said “Refs:Glob Chang Biol. 2014 Aug;20(8):2450-8. doi: 10.1111/gcb.12617. Epub 2014 May 26. Water temperature and fish growth: otoliths predict growth patterns of a marine fish in a changing climate. Adam N Rountrey 1, Peter G Coulson, Jessica J Meeuwig, Mark Meekan The effect of temperature on growth of Atlantic cod (*Gadus morhua* L.) K. M. Brander *ICES Journal of Marine Science*, Volume 52, Issue 1, February 1995, Pages 1–10, [https://doi.org/10.1016/1054-3139\(95\)80010-7](https://doi.org/10.1016/1054-3139(95)80010-7)

Punt, A.E., Amar, T., Bond, NA, Butterworth, DS, de Moor, CL, DeOliviera, JAA et al. 2014. *ICES Journal*, 71(8), 2208-2220. Fisheries management under climate and environmental uncertainty, control fits and performance simulation.

Beasell-Browne, P., Punt, AE, Tuck, GN, Day, J, Klaer, N, Penney, A. *Fisheries Research* 106306, 2022. The effects of implementing a dynamic B0 harvest control rule in Australia’s Southern and Eastern scalefish and shark fishery.

Amar, ZT, Punt AE, Dorn, MW. 2009. The evaluation of two management strategies for the Gulf of Alaska walleye pollock fisheries under climate change, *ICES Journal of Marine Science* 66(7), 1614-1632

Punt AE, Szuwalski, CS, Stockhausen, W. 2014. An evaluation of shark-recruitment proxies and environmental change points for implementing the US sustainable fisheries act, *Fisheries Research*,

157, 28-40.

Thorpe RB, Arroyo, NL, Safi, G., et al. 2022. The response of North Sea Ecosystem functional groups to warming and changes in fishing, *Frontiers in Marine Science*, 9:841909.

Andersen, 2020. *Fish Ecology, Evolution and Exploitation*. Princeton.

Pope et al., 2020, Scrabbling around for understanding of natural mortality, *Fisheries Research*, 240, 105952.

Caswell et al., 2020. Something old, something new: Historical perspectives provide lessons for blue growth agendas, *Fish and Fisheries*.

Worm, B., Hilborn, R., Baum, J. K., Branch, T. A., Collie, J. S., Costello, C., & Fogarty, M. J. (2009). Rebuilding global fisheries. *Science*, 31, 578– 585.

Thorpe, R. B., & De Oliveira, J. A. A. (2019). Comparing conceptual frameworks for a fish community MSY (FCMSY) using management strategy evaluation – an example from the North Sea. *ICES Journal of Marine Science*. <https://doi.org/10.1093/icesjms/fsz015>.

Mesnil B. (2012). The hesitant emergence of maximum sustainable yield (MSY) in fisheries policies in Europe. *Marine Policy*, 36, 473–480”

All references added and discussed in the revision.

Reviewer 1 said “Annex: Caveats associated with the SP model approach.

Having a spatial network of SP models is novel, but – need to caveat:

a) No size structure

b) $dr/dT = 0$ assumption (climate impacts only through k)

c) $r=0.5$ assumed for cod.

d) MSY at $B_0 / 2$ despite evidence to the contrary.

e) Assumptions about the nature of species niches (compare with other model approaches).

f) What assumptions are made about migration between SP regions? It seems the stock is assumed static.

g) Stock assumed independent of other stocks/fisheries.”

We have added a section devoted to the potential caveats of our approach lines 305 page 10 (see our above answer to the reviewer).

Reviewer #2 (Remarks to the Author):

Reviewer 2 said “General comments. This paper outlines a model to try to disentangle attribution of fisheries vs climate influenced environmental drivers on stock biomass trajectories through time – both historical time series and under CMIP6 projections. The application of the approach is to cod in the North Sea. The work is interesting but requires more links to existing literature and the current directions of fisheries management/science. Also, I think the paper requires more justification of decisions made during the modelling process. I have also included some other things I think it would be good to address in the specific comments.”

We have added more than 50 new references and have improved the discussion section. We have also more carefully discussed the choice of threshold values. In particular, thresholds for collapse is now 0.1 for all analyses, a threshold that is more frequently found in the literature than 0.05. This does not affect our results. Below we answer in details to the specific comments.

Reviewer 2 said “1. Throughout the manuscript: please replace ‘scenarii’ with ‘scenarios’. Scenarii is technically correct, but the use of the more common term scenarios might be kinder on the general readership.”

We have changed scenarii by scenarios throughout the paper.

Reviewer 2 said “2. Line 14: While I can live with the use of the acronym CIEC it is worth considering throughout this paper whether the use of acronyms (including dSSB and mdSSB) help the less familiar reader understand what you have been doing.”

We think we need to keep these different terms because they are essential in the paper. To consider the reviewer comment however, we have moved Supplementary Table 1 in the main text. By this way the reader will see more rapidly and directly the meaning of each acronym.

Reviewer 2 said “3. Lines 22-23: I think would read more smoothly as “... fisheries management immediately adjusts fishing effort...”

Modified directly in the revision.

Reviewer 2 said “4. Line 23, ‘effort, will therefore create a deleterious response lag’: Is it possible to add a sentence on why it is deleterious here (even if covered more fully in the main text)?”

We now say line 23 in the abstract:

“Failure to monitor CIEC, so that fisheries management immediately adjusts fishing effort in response to environmentally-driven shifts in stock productivity, will therefore create a deleterious response lag that may cause the stock to collapse.”

Reviewer 2 said “5. Lines 35-37: I do not agree with the statement that the contributions and interaction of fisheries and climate change are unknown. There has been a lot of attention paid to this over the past decade - work in the Baltic (e.g. by Möllmann at Uni Hamburg) is just one example. Often model based due to the scarcity of sufficient and reliable time series for both drivers.”

We agree. We have modified the sentence in the revision and we have added a few citations. We say **lines 38-42 page 3**:

“Although many studies have investigated how fishing and environment may interact to affect a fish stock^{42, 43, 44}, the precise respective contribution of fishing and CIEC and how this varies in time remains poorly known, yet this knowledge is likely to be fundamental to effective fisheries management⁴⁵.”

Reviewer 2 said “6. Lines 46-48: Use of observed spatial distributions as the validation of the ‘no fishing’ case is perhaps not the best test given we only have observations resulting from a combination of the two.”

We agree. That is why we have examined all cases and scenarios in our paper (Figures 2-4).

Reviewer 2 said “7. Line 67: Why is a relationship between fishing intensity and F remarkable? Isn't it a fundamental assumption in fisheries and what you are doing?”

“Remarkable” has been removed from the revision.

Reviewer 2 said “8. Line 91: Given what you based the cluster analysis on I would say “... performed on long-term reconstructed changes in...”

We have added “reconstructed” in the revision.

Reviewer 2 said “9. Line 107: I think you need to spell out what you think the dichotomy is clearly fairly early on in the document”

We agree. We now clarify what we mean by “dichotomy” **lines 285-287 page 9**:

“Our results therefore emphasise how both fishing and climate must be considered to resolve the apparent dichotomy (i.e. the debate between the respective contribution of fishing and environment on a fish stock) they create for fisheries management⁴⁵.”

Reviewer 2 said “10. Lines 124-125 ‘collapse and fishing effort reduced to ~34-36% from 2008’: The reduction refers to the contribution by fishing not a 34-36% drop in actual effort right? (Ambiguous as written)”

Yes. We have clarified this point in the revision **line 150 page 5**:

“A reduction in fishing intensity due to fish management allowed the stock to avoid collapse and fishing effort reduced to reach values of ~34-36% from 2008 (P7).”

Reviewer 2 said “11. Line 132: How does the ESM driven estimates compare to your reconstruction? Any consistency to historical gross pattern? I think you can explore this set of results in more depth.”

We already say in the main text lines 159-163 page 6:

“Although our estimates showed pronounced inter-ESM variability for both emission scenarios (i.e. thin black curves and average in thick green for 1850-2019, thin dashed blue and red curves for 2020-2100 for scenarios SSP245 and SSP585, respectively), future mdSSB (i.e. with no fishing) were predicted to decrease substantially during the forthcoming century (Fig. 3a, thick full blue and red curves for the average of all SSP245 and SSP585 scenarios, respectively).”

However, we have added a new sentence in the discussion to consider the inter-ESM variability. We say lines 361-365 page 11 of the revision:

“Inter-ESM variability remains important and it is clear that this affects our projections (Fig. 3-5). In addition, emission scenarios are inherently unpredictable and this might also influence our projections, although in more expected ways (Fig. 4-5). However, the model we propose could be used on a year-to-year basis to better anticipate future changes in SSB and predict more realistic fishing quotas that may either prevent stock collapse or better optimise exploitation.”

Reviewer 2 said “12. Line 145-146: You explore the outcome of a constant catch case, but that isn’t particularly realistic in terms of responses in the region. So what if time varying F (based on stock status) was used, can the stocks survive?”

We agree. This is what we have done in the analyses that give Figures 4-5.

Reviewer 2 said “13. Lines 160-164: What if you allow rebuilding to high SSB (like a B or even A point) as part of the response to climate so you then have more control via fisheries management lever alone? Is that possible/feasible?”

We have clarified this point in the revision line 179-191 page 6 (see the underlined parts):

“mdSSB (ordinate on Fig. 3b) denotes the maximum dSSB achievable for a given environmental regime; dSSB is always below mdSSB. Expectedly, alleviating fishing effort is the only way to maintain a stable SSB when the environmental regime becomes less suitable⁴⁶. Although it is possible to maintain cod SSB when the environment is highly suitable, such as Nordic seas (e.g. for $K_t > 0.5$ in Fig. 3b), it is harder, if possible, to achieve in the environmentally less favourable North Sea (Fig. 3b, see also Fig. 1a). This can be illustrated by the three points A, B and C in Fig. 3b. For a hypothetical dSSB corresponding to point A, we see that increasing dSSB by fish management (i.e. along the horizontal line from the starting point A to the left) is easier than for a dSSB corresponding to points B and C (Fig. 3b); this is because the number of isolines to the left of each point, reflecting the scope to reduce fishing intensity, decreases from A to C. At point C, it becomes nearly impossible to keep dSSB stable by cod management because along the horizontal line from the starting point C to the left the number of isolines is considerable reduced.”

Reviewer 2 said “14. Lines 165-169: Could people use the overwhelming influence of the climate driver to make the argument “fish it down as going anyway”?”

Yes that could be the case in the North Sea at the end of the century.

Reviewer 2 said “15. Paragraph 170-187, especially line 186-187 ‘In addition, our results suggest that preventing collapse is easier than trying to reverse a collapse.’: This is particularly true if you are trying to rebuild to a level that is no longer possible under your new environmental regime. I think you need to reflect on this point so more. The use of fixed reference points is a little behind where the thinking of fisheries management agencies in developed world fisheries so it would be good to see the paper address the more dynamic viewpoint management and fisheries science is moving to.”

We thank the reviewer for this key point. We have added a few sentences at the end of the paragraph. We say line 301-303 page 10:

“This is particularly true if managers try to rebuild to a level that is no longer possible under a new environmental regime. These findings also show how important is to manage fish stocks using dynamic reference points²⁰.”

Reviewer 2 said “16. Lines 203-204 ‘Although it was impossible to strictly compare the influence of fishing and CIEC because fishing intensity was hypothetical in our scenarii’: Why? Being hypothetical does not prevent you applying the method.”

We agree. We have improved the sentence. We now say **line 235-235 page 7**:

“Although fishing intensity was hypothetical in our scenarios of changes, the analysis clearly suggests that both drivers are important to consider in future projections.”

Reviewer 2 said “17. Lines 225-228: This statement is something that I think is not a new realisation for many in fisheries science...”

We agree but some papers continue to be published with this view. We can cite Brander⁴⁷ who has also worked on the implications of climate in fisheries and Cardinale & Svedäng⁴⁸. They are cited in our revision.

Reviewer 2 said “18. Lines 400-401: What does ‘0.1 for display purpose, starting by ESMs based on scenario SSP245 followed by scenario SSP585’ mean? I think the caption should also read ‘...recovery of the stock to a target standardised...’”

We have clarified this point in the figure legend **line 763 page 21** (underlined):

“Fishing intensity, unpredictable for 2020-2100, was fixed to be arbitrarily constant between 0.08 and 0.17 by increment of 0.1 for display purpose (i.e. high resolution of the colour diagram), starting by ESMs based on scenario SSP245 followed by scenario SSP585.”

The second point has been modified directly in the revision.

Reviewer 2 said “19. Lines 403-406: This means you are judging your target reference point vs some historical max mdSSB not the current climate influenced state? To be more reflective of where fisheries science is/is going it would be good to see what happened if you adjusted the target dSSB as the mdSSB drops with climate?”

We agree and the analysis suggested by Reviewer 2 was carried out in Figures 4-5.

Reviewer 2 said “20. Lines 456-457 ‘An annual index is then calculated by standardised principal component analysis based on the period March-September’: To make this paper standalone, this method needs to be repeated here or in the supplementary materials.”

We now describe fully the procedure to create the plankton index in the revision **lines 819-834 page 22**:

“We used a plankton index of larval cod survival, which was an update of the index proposed by Beaugrand and colleagues³¹. Based on data from the Continuous Plankton Recorder (CPR)⁴⁹, the index is based on the simultaneous consideration of six key biological parameters for the diet and growth of cod larvae and juveniles in the North Sea^{50, 51}: (i) Total calanoid copepod biomass as a quantitative indicator of food for larval cod, (ii) mean size of calanoid copepods as a qualitative

indicator of food, (iii-iv) the abundance of the two dominant congeneric species *Calanus finmarchicus* and *C. helgolandicus*, (v) the genus *Pseudocalanus* and (vi) the taxonomic group euphausiids. A standardised Principal Component Analysis (PCA) is performed on the six plankton indicators for each month from March to September for the period 1958-2017 (table 60 years x 7 months-6 indicators). The plankton index is simply the first principal component of the PCA³¹. “

Reviewer 2 said “21. Line 477: Explain why ‘It is important to note that α should not be mistaken with ICES fishing effort F .’”

We have clarified this point in the revision lines 854-858 page 23:

“ α is the fishing intensity that varies between 0 (i.e. no fishing) and 1 (i.e. 100% of SSB fished in a year). It is important to note that α (see Equation 10 below) should not be mistaken with ICES fishing effort F ⁴¹ (calculated from SSB). The second term of Equation 1 is the intrinsic growth rate of the fish stock that is a function of both K_t and the population growth rate r (r was fixed to 0.5 in most analyses, but see Fig. 3d however where r varied from 0.25 to 0.75).”

Reviewer 2 said “22. Line 479: Why would r be fixed under climate forcing?”

We have clarified this point in the revision lines 859-864 page 23:

“The population growth rate r is highly influenced by the life history traits of a species²⁸ but also by environmental variability^{4, 5, 29}. Here, the population growth rate was assumed to be constant in space and time and the influence of environmental variability occurred exclusively through its effects on K_t . We made this choice to not multiply the sources of complexity and errors (i.e. population growth rate is very difficult to assess and varies with age²⁸). The third term reflects the part of $dSSB$ that is lost by fishing.”

Reviewer 2 said “23. Lines 487-488: If you don’t have chlorophyll a data for the entire time period, how did you patch the gap for the longer time series?”

We have clarified this point in the revision lines 813-815 page 22:

“These data were only used to map the average maximum standardised SSB (mdSSB) around the North Sea (Fig. 1a). When long-term changes in mdSSB were examined, we used modelled chlorophyll data (see section “Climate projections” below).”

We also say lines 846-847 page 23:

“Long-term changes in modelled SSB were based on these data (including modelled daily chlorophyll data).”

We have also clarified the legend of Figure 1.

Reviewer 2 said “24. 495-496: Please justify the decision around why the thermal niche matched is a good choice.”

We have added this Supplementary Figure (Supplementary Figure 2) in the revision.

Supplementary Figure S2. Histogram of the number of geographical cells with a cod occurrence as a function of sea surface temperature (blue bar) from Beaugrand and colleagues²⁴ and the thermal response curve chosen in this study (red).

We also say lines 881-885 page 24:

“Supplementary Fig. 2 compares the thermal response curve we chose in the present study with the data analysed in Beaugrand and colleagues²⁴. The figure shows that the response curve (red curve) is close to the histogram showing the number of geographical cells with a cod occurrence as a function of temperature varying between -2°C (frozen sea water) and 20°C.”

Reviewer 2 said “25. Line 499-500: What does “not very much” mean here? A quantitative stat used or was it heuristic?”

We have clarified this point in the revision lines 888-890 page 24:

“Slight variations in the different parameters of the niche did not alter either the spatial patterns in the distribution of mdSSB nor the correlations with recruitment.”

Reviewer 2 said “26. Line 502: Please explain why the bathymetric niche can’t be Gaussian or delete this sentence and rely on the closing sentence of the paragraph (on frequent use of this approach).”

We have clarified this point in the revision lines 891-903 page 24:

“To model the bathymetric niche of cod, we used a trapezoidal function. Changes in mdSSB, U_2 , along bathymetry, were assessed using four points ($\vartheta_1, \vartheta_2, \vartheta_3, \vartheta_4$):These parameters were retrieved from the literature^{52, 53}.”

Reviewer 2 said “27. Line 514-516 (trophic niche description): Please explain why this approach was taken”

We have clarified this point in the revision lines 904-909 page 24:

“The trophic niche was modelled by a rectangular function on a daily basis. To our knowledge, no information on the trophic niche is available. We modelled the trophic niche by fixing U_3 to 1 when

chlorophyll-a concentration was higher than 0.05 mg.m⁻³ during a minimum period of 15 days and 0 otherwise (Supplementary Fig. 1). This minimum of chlorophyll was implemented as a proxy for suitable food, which has been shown to be important in the North Atlantic for cod recruitment and distribution^{26, 31}.

Reviewer 2 said “28. Line 519 (equation 9): What is the justification for straight multiplication? With no weighting? It's fine as any starting assumption but you need to explain why you chose it.”

We have added the following text in the revision **lines 910-916 page 24**:

“There exists two ways to combine the different ecological dimensions of a niche: (i) use an additive or (ii) a multiplicative model^{30, 40}. We have used a multiplicative model because if one dimension is associated to a nil abundance, the resulting abundance combining all dimensions will be nil in contrast to an additive model; therefore only one unsuitable environmental value may explain a nil abundance. All dimensions were associated to abundance values that vary between 0 and 1³⁰.”

Reviewer 2 said “29. Line 525-526: Any attempt to check that climatology vs local datasets, as Chl is likely to have been influenced by climate change so is climatology over that period of time a good choice? Was use of a reanalysis product considered?”

We have clarified this point in the revision. The climatology was exclusively done to map the average spatial distribution mdSSB. Modelled chlorophyll daily data were subsequently used for all other analyses. See our changes in the revision **lines 813-816 page 22**:

“These data were only used to map the average maximum standardised SSB (mdSSB) around the North Sea (Fig. 1a). When long-term changes in mdSSB were examined, we used modelled chlorophyll data (see section “Climate projections” below).”

We also say **lines 846-847 page 23**:

“Long-term changes in modelled SSB were based on these data (including modelled daily chlorophyll data).”

We have also clarified the legend of Figure 1.

Reviewer 2 said “30. Lines 540-544 (standardisation of ICES SSB): I would like this explained more fully and transparently. How were the standardisations done? Perhaps also tabulate the correlations.”

We have improved this section. We now say **lines 948-956 page 25**:

“Standardisation of ICES SSB, necessary for this analysis, was complicated because many different kinds of standardisation were achievable so long as X remained strictly above 0 (i.e. full cod extirpation, not observed so far¹⁹) and strictly below $\min(K)$ (i.e. all black curves always below all points of the blue curve were possible, Supplementary Fig. 2). Indeed, ICES SSB includes exploitation and environmental fluctuations whereas K (i.e. mdSSB) integrates only environmental forcing; the difference is mainly caused by the negative influence of fishing. We chose the black curve (ICES SSB) that maximised the correlation between α (fishing intensity in the FishClim model) and F (ICES fishing effort)¹⁹.”

Reviewer 2 said “31. Lines 548-551: Why not time varying? Is there information on fleet size etc that could be an independent guide?”

First, we considered variation in the environment and we fixed fishing intensity. Second, we fixed the environment and we considered variation in fishing intensity. Therefore, we tested both influences. We have clarified this point in the revision lines 960-961 page 26:

“First, we fixed fishing intensity and considered exclusively environmental variations through its influence on dSSB.”

Reviewer 2 said “32. Lines 551-554: Again, why not allow for time varying?”

See also our answer above. We have clarified this point in the revision line 965-966 page 26:

“Second, we fixed the environmental influence on dSSB and considered variations in fishing intensity.”

See also Fig. 3b-c that shows how both environmental influences and changes in fishing intensity may affect SSB simultaneously.

Reviewer 2 said “33. Line 604: Why was 0.05 used as an extirpation threshold?”

We have homogenised the threshold to 0.1 everywhere in the paper (see our answer to Reviewer 1 comment). We have redone our analysis for Fig. 3d using a threshold of 0.1 and our interpretation for Fig. 3a-c uses a threshold of 0.1. Note this does not alter our conclusions. We also provide new references in the revision to justify this choice.

Reviewer 2 said “34. Line 607-609: Please explain how the superposition allows for an analysis of fishing and environmental interactions.”

We have rewritten the sentence as follows, lines 1022-1023 page 27:

“We modelled dSSB as a function of fishing intensity α and CIEC to show how fishing and the environment interact (Fig. 3b-c).”

We also better explain these results in the main text lines 177-189 page 6:

“mdSSB (ordinate on Fig. 3b) denotes the maximum dSSB achievable for a given environmental regime; dSSB is always below mdSSB. Expectedly, alleviating fishing effort is the only way to maintain a stable SSB when the environmental regime becomes less suitable⁴⁶. Although it is possible to maintain cod SSB when the environment is highly suitable, such as Nordic seas (e.g. for $K_t > 0.5$ in Fig. 3b), it is harder, if possible, to achieve in the environmentally less favourable North Sea (Fig. 3b, see also Fig. 1a). This can be illustrated by the three points A, B and C in Fig. 3b. For a hypothetical dSSB corresponding to point A, we see that increasing dSSB by fish management (i.e. along the horizontal line from the starting point A to the left) is easier than for a dSSB corresponding to points B and C (Fig. 3b); this is because the number of isolines to the left of each point, reflecting the scope to reduce fishing intensity, decreases from A to C. At point C, it becomes nearly impossible to keep dSSB stable by cod management because along the horizontal line from the starting point C to the left the number of isolines is considerable reduced.”

Reviewer 2 said “35. Line 656: Please be clear here, you intentionally caused extirpation by $F > F_{msy}$ as you want to see how the environment hastens it?”

No. The reviewer misunderstood our point. We have clarified the sentence in the revision lines 1077-1079 page 29:

“We used this concept to show that controlling fishing intensity delayed cod extirpation. From Equation 1, we calculated fishing intensity, called α_{MSYt} , so that X remained above X_{MSYt} at all time t :”

Reviewer 2 said “36. Line 667: Be clear on why it is being called ‘standardised catch’ (because the K is 0-1 so the catch is also automatically standardised? Or for a different reason?)”

We agree. We have clarified the sentence line 990 page 26. We now say:

“In term of fishing exploitation, we assessed pooled standardised catch (i.e. pooled dSSB) in 2100 (2020-2100), again for two scenarios of CIEC (SSP245 and 585) and two scenarios of cod management (constant versus adjusted -MSY- fishing intensity; Fig. 5).”

We have also clarified the goal of this analysis lines 1090-1096 page 29:

“The goal of this analysis was to demonstrate that controlling fishing intensity optimises cod exploitation.”

Reviewer 2 said “37. Line 670-671: This should be ‘Finally, we assessed the median of the percentage of reduction...”

We have added the “the” in the revision.

Reviewer 2 said “38. Table 1: Note that mdSSB is also constrained between 0 and 1. Also you misspelt logarithm for the ICES SSB row”

We have made the changes in the revision (Table 1).

Reviewer 2 said “39. Figure 1: Any possibility of using colours more friendly to colour blind eyes?”

Yes we certainly can. See the new Fig. 1. We have replaced the green/blue curves by red/blue curves in panel c.

Reviewer 2 said “40. Figure 1b: The relationship with recruitment does seem reliable and is plausible, but what is the risk it is an ephemeral relationship as seen in previous published recruit relationships that are environmentally conditioned? It looks to be largely useful for 1965-2015 but what about more recently? Any commentary on uncertainty going forward?”

It is difficult to say and we do not want to speculate here. Therefore, we do not think we should mention this possibility.

Reviewer 2 said “41. Figure 1f: Please use different patterns or colours for each line please (as not simply confidence ranges but represent different things). Also, is it possible to include a distance metric to conclusively show that “closer” to good conditions early on and “poor conditions” later? Easy to see it about 2005, but more difficult in other years.”

We now separate the upper and lower curves and have added horizontal lines to enhance readability. See revised Fig. 1f.

Reviewer 2 said “42. Figure 2b: Why is environment only a positive influence? Or just referring to the direction in the plot?”

We have improved the figure. See the negative and positive environmental influences on Fig. 2b.

Reviewer 2 said “43. Figure 4: The magenta line is very hard to actually see. Your description of what the histograms are showing could be clearer I think. Perhaps describe it as a frequency histogram of time to extirpation (counted per grid cell) in each comparison or something along those lines. Also why was only the magenta area used for the calculation not the entire modelled area?”

We have changed the colour magenta lines to grey/pink lines in the revision (see Figures 4 and 5). We think it is better.

We have modified the figure legend. We now say:

“Frequency histograms of difference between maps of time to extirpation for the North Sea (51°N-62°N and 3°W-9.5°E).”

As we say in the main text, we focussed on the North Sea because ICES recruitment and SSB as well as the plankton index originated from the North Sea.

Reviewer 2 said “44. Figure 5: Using the word ‘reduction’ instead of ‘diminution’ might be better”

We changed it in the revised figure.

Reviewer 2 said “45. Supplementary Figure 2: Should this be ‘... ICES Spawning Stock Biomass (SSB) should be at or below any point of the maximum dSSB... ’? Also is it possible to not correlation coefficients on the curves so its clear how the red line maximises the correlation and what the variation in the correlation was across the lines.”

We have made the correction in the figure legend.

We think that adding the correlation on each line is not really possible. They are too many.

Reviewer #3 (Remarks to the Author):

Reviewer 3 said “The topic is important and of general interest and your analyses seems to be rigorous and solid.”

We agree.

Reviewer 3 said “However, I have severe problems with understanding the whole background for your paper and find the way it is structured confusing.”

We have clarified the Methods section in the revision and thank the reviewer for his comments that help to improve the ms. We specifically answer to her/his comments below.

Reviewer 3 said “I also find the level of precision to be low many places, leaving me unsure about what you mean.”

We have carefully considered all comments made by the reviewer below.

Reviewer 3 said “Two main issues: The title, Fisheries management, solving the dichotomy of fishing and climate, and what you write implies that there is an ongoing discussion on EITHER fishing or climate-induced environmental variability determining fluctuations in exploited fish stocks. While this may have been a discussion several decades back numerous papers and work in ICES WGS and other places has led to a (more or less) consensus that both these two factors must be considered. It’s for sure discussed to what degree each factor contributes, this may vary in time and between stocks, but not either/or.” Your FishClim model is likely a very good way of inspecting the influence of fishing vs environmental factors, but the either/or setting is not valid.”

We disagree. We provide two examples:

Brander wrote in 2018⁴⁷ *“Climate change not to blame for cod population decline”*. The content of the article published in Nature Sustainability is clear. He says in the abstract: *“Three decades of increasing temperature were expected to cause cod to decline in the North Sea and Gulf of Maine, but the stocks increased in the former and declined in the latter area. These trends are due to changes in fishing pressure rather than climate change.”*

Another example. Cardinale & Svedäng (2011)⁴⁸ in MEPS wrote *“The recovery of the cod stock during a ‘cod-hostile’ ecological regime indicates that fisheries are the main regulator of cod population dynamics in the Baltic Sea.”*

These two examples show that this debate remains nowadays.

Nevertheless, we have modified the last sentence of the abstract:

“Consequently, the application of FishClim, which quantifies the respective influence of fishing and climate, will help to develop better strategies for sustainable, long-term, fish stock management.”

And in the conclusion:

“Although this dichotomy has waned over time and that more and more studies are considering the influence of the two drivers^{42, 43, 44, 54}, this dichotomy regularly reappears^{47, 48}.”

Reviewer 3 said “Your FishClim model is likely a very good way of inspecting the influence of fishing vs environmental factors, but the either/or setting is not valid.”

We agree.

Reviewer 3 said “I find the structure of your ms very confusing. It’s OK to depart from standard format, but I do not understand why you already from early on mix Introduction, Results, and Discussion.”

This paper was submitted to Nature Communications. We have restructured the paper accordingly.

Reviewer 3 said “Some central concepts are not introduced very well. After reading the whole ms I’m still unsure about what CIEC actually is. The general concept is OK, although you introduce it without a clear description or reference. However, the main problem is that it’s not obvious which variables it represents here. Sea temperature (in a fixed location or gridded), phytoplankton, wind direction? OK to have it at a conceptual level first, but when you get down to a concrete cod stock readers want to know clearly what the actual environmental drivers are.”

We have clarified the term CIEC in the revision. We have moved Supplementary Table 1 in the main text (see Table 1). This table explains better what CIEC means. In this table, we define CIEC as follows:

“All environmental alterations that result from climatic variability and anthropogenic climate change. In this paper, we considered changes in sea surface temperature, chlorophyll-a concentration and a sliding 15-day period above a chlorophyll-a concentration level of 0.05 mg.m⁻³.”

Therefore, in the introduction we say **line 36 page 3**:

“Managing fish stocks has always been a difficult task because stocks exist in complex ecosystems that can experience substantial changes triggered by extrinsic (e.g. fishing and CIEC, see definition of CIEC in Table 1) and intrinsic (e.g. biological or ecological processes) forces^{3, 26, 55}.”

Reviewer 3 said “16. Depends on what CIEC is. Many environmental time series (like sea temperature or the NAO) go further back in time than commercial, industrial fisheries.”

We have clarified what we mean by CIEC in our answer above.

Reviewer 3 said “48. There is unclarity to when you are describing fish in general and when a specific cod stock. Cod is not mentioned (except in the Abstract) before here. It’s normal scientific practice to introduce a species also with its latin name. You write about the North Sea cod stock or all cod in the North Sea (including various coastal populations)?”

We have clarified this point in the revision by adding a paragraph **lines 43-52 page 3**:

*“The Atlantic cod *Gadus morhua* L. has declined in the North Sea since the end of the gadoid outburst⁵⁶ and there has been a debate on whether or not CIEC has contributed with overfishing to the diminishing Spawning Stock Biomass (SSB)^{25, 26, 47, 48}. Surprisingly, although some studies have jointly investigated the influence of CIEC and fishing on cod SSB^{26, 42, 57}, there have been no attempt to quantify precisely the effects of the two drivers despite their importance in term of stock management. As a result, current management practices continue to ignore the potential influence of CIEC on cod stocks⁵⁷. This is especially worrying since anthropogenic climate change is having a discernible influence on many marine ecosystems and that its impacts may drastically increase in the decades to come^{58, 59 60, 61, 62}.”*

Reviewer 3 said “44, 49 ++ The geographic extent of your study is unclear. “Northeast Atlantic” (huge area), “seas around the UK” , “North Sea” ??”

We have clarified this point in the introduction section (see also the new paragraph above). We have also made a few changes in the first and third paragraphs of the introduction section.

Reviewer 3 “said “50 + this is a mix of method description, results, and discussion. Very confusing.”

We have restructured the paper so that it is structured in a classical way.

Reviewer 3 “53 What is “observed ICES SSB”? How can SSB be observed? Is this the VPA-based SSB? The reference, num 21, is difficult to check based on the information given.

We agree. This point has been clarified in the revision line 935 page 25. We now say ‘ICES-based SSB’.

Reviewer 3 “67 Why is this remarkable, isn’t it to be expected?”

We have removed the word “remarkable” from the revision (see line 92 page 4).

Reviewer 3 “75 Discussion hard to follow, as we don’t know which environmental variables are captured in CIEC”

The acronym CIEC has been clarified in Table 1. See our previous answer on this point.

Reviewer 3 “128 + this is likely correct, but attribution to anthropogenic CC (as opposed to natural fluctuations) is still difficult and was for sure not possible when the papers referred to where written (the only recent one, Brander et al., is a review and says nothing about the changes being anthropogenically driven)”

The reviewer is correct. We have modified the sentence in the revision lines 154-159 page 6:

“Climate change (natural and/or anthropogenic) has affected the environment of the North Sea by altering plankton composition and ecosystem trophodynamics^{31, 63, 64}. We forced our model by outputs from four Earth System models (ESMs) based on two scenarios of SST/Chlorophyll changes (i.e. Shared Socio-economic Pathways SSP245 and SSP585, Methods) to assess mdSSB (K_t in Equation 1) for the period 1850-2100 and examined the potential influence of anthropogenic climate change.”

Reviewer 3 “153 Nordic Seas? Where is this and which cod stock are you writing about? This area is normally taken to be the Greenland Sea, Norwegian Sea, and Iceland Sea, a deep-sea region not suited for cod.”

We have clarified this point in the revision lines 182-184 page 6:

“Although it is possible to maintain cod SSB when the environment is highly suitable, such as the Icelandic cod stocks (e.g. for $K_t > 0.5$), it is harder, if possible, to achieve in the environmentally less favourable North Sea (Fig. 3b and Fig. 1a).”

Reviewer 3 “167-169 This is for sure correct, but very much an understatement. With the worst case SSP585 there will, with very high confidence, not be cod in the North Sea.”

We agree but we prefer to keep our statement as it is because some individuals might still be found in the North Sea. We consider it is too extreme to say that.

Reviewer 3 “225 yes, when the book was written the dichotomy existed. Fortunately, the either/or discussion was abandoned some years later”

We disagree. We provide two examples:

Brander wrote in 2018⁴⁷ “*Climate change not to blame for cod population decline*”. The content of the article published in Nature Sustainability is clear. He says in the abstract: “*Three decades of increasing temperature were expected to cause cod to decline in the North Sea and Gulf of Maine, but the stocks increased in the former and declined in the latter area. These trends are due to changes in fishing pressure rather than climate change.*”

Another example. Cardinale & Svedäng (2011)⁴⁸ in MEPS wrote “*The recovery of the cod stock during a 'cod-hostile' ecological regime indicates that fisheries are the main regulator of cod population dynamics in the Baltic Sea.*”

Reviewer 3 “235- this seems so obvious that I don’t see the point”

We think this statement should be kept in the revision because this is again another example that shows how important it is to mitigate anthropogenic climate change.

Reviewer 3 “242 Totally agree, but this is not a new idea”

We have slightly modified this point in the revision lines 384-388 page 11:

“*Although our analysis focused on North Sea cod because of the depth of understanding of this fishery and the comprehensive data available, we expect our findings to be widely applicable and so we encourage a better consideration of fishing and CIEC in all future fisheries management.*”

References used in our answers

1. Punt AE, Huang T, Maunder MN. Review of integrated size-structured models for stock assessment of hard-to-age crustacean and mollusc species. *ICES Journal of Marine Science* **70**, 16-33 (2013).
2. Pope JG, Gislason H, Rice JC, Daan N. Scrabbling around for the understanding of natural mortality. *Fisheries Research* **240**, 105952 (2021).
3. Jennings S, Kaiser MJ, Reynolds JD. *Marine Fisheries Ecology*. Blackwell Science Ltd. (2001).
4. Rountrey AN, Coulson PG, Meeuwig JJ, Meekan M. Water temperature and fish growth: otoliths predict growth patterns of a marine fish in a changing climate. *Global Change Biology* **20**, 2450-2458 (2014).

5. Brander K. The effect of temperature on growth of Atlantic cod (*Gadus morhua* L.). *ICES Journal of Marine Science* **52**, 1-10 (1995).
6. Pethybridge H, Roos D, Loizeau V, Pecquerie L, Bacher C. Responses of European anchovy vital rates and population growth to environmental fluctuations: An individual-based modeling approach. *Ecological Modelling* **250**, 370-383 (2013).
7. Huserbraten MBO, Moland E, Albretsen J. Cod at drift in the North Sea. *Progress in Oceanography* **167**, 116-124 (2018).
8. Daan N. Changes in cod stocks and cod fisheries in North Sea. *Rapports et Procès-verbaux des Réunions du Conseil International pour l'Exploration de la Mer* **172**, 39-57 (1978).
9. Righton D, Quayle VA, Hetherington S, Burt G. Movements and distribution of cod (*Gadus morhua*) in the southern North Sea and English Channel: results from conventional and electronic tagging experiments. *Journal of the Marine Biological Association of the United Kingdom* **87**, 599-613 (2007).
10. Righton D, Metcalfe J, Connolly P. Different behaviour of North and Irish Sea cod. *Nature* **411**, 156 (2001).
11. Hutchinson WF, Carvalho GR, Rogers SI. Marked genetic structuring in localised spawning populations of cod *Gadus morhua* in the North Sea and adjoining waters as revealed by microsatellites. *Marine Ecology Progress Series* **223**, 251-260 (2001).
12. Robichaud D, Rose GA. Migratory behaviour and range in Atlantic cod: inference from a century of tagging. *Fish and Fisheries* **5**, 185-214 (2004).
13. Larkin PA. An epitaph for the concept of Maximum Sustained Yield. *Transactions of the American Fisheries Society* **106**, 1-11 (1977).
14. Kempf A, *et al.* The MSY concept in a multi-objective fisheries environment – Lessons from the North Sea. *Marine Policy* **69**, 146-158 (2016).
15. Punt AE, Szuwalski C. How well can FMSY and BMSY be estimated using empirical measures of surplus production? *Fisheries Research* **134-136**, 113-124 (2012).
16. Mesnil B. The hesitant emergence of maximum sustainable yield (MSY) in fisheries policies in Europe. *Marine Policy* **36**, 473-480 (2012).
17. Thorpe RB, De Oliveira AA. Comparing conceptual frameworks for a fish community MSY (FCMSY) using management strategy evaluation—an example from the North Sea. *ICES Journal of Marine Science* **76**, 813-823 (2019).

18. Punt AE, Szuwalski CS, Stockhausen W. An evaluation of stock-recruitment proxies and environmental change points for implementing the US sustainable Fisheries Act. *ICES Journal of Marine Science* **157**, 28-40 (2014).
19. ICES. Cod (*Gadus morhua*) in Subarea 4, Division 7.d, and Subdivision 20 (North Sea, eastern English Channel, Skagerrak). In: *Report of the ICES Advisory Committee, 2021. ICES Advice 2021, cod.27.47d20* (2021).
20. Bessell-Browne P, Punt AE, Tuck GN, Day J, Klaer N, Penney A. The effects of implementing a 'dynamic B0' harvest control rule in Australia's Southern and Eastern Scalefish and Shark Fishery. *Fisheries Research* **252**, 106306 (2022).
21. Worm B, *et al.* Rebuilding Global Fisheries. *Science* **325**, 578-585 (2009).
22. Schickele A, *et al.* Modelling European small pelagic fish distribution: methodological insights. *Ecological Modelling* **416**, 108902 (2020).
23. Schickele A, *et al.* Redistribution of small pelagic fish in Europe and Climate Change. *Fish and Fisheries* **22**, 212-225 (2021).
24. Beaugrand G, Lenoir S, Ibanez F, Manté C. A new model to assess the probability of occurrence of a species based on presence-only data *Marine Ecology Progress Series* **424**, 175-190 (2011).
25. Beaugrand G, Kirby RR. Spatial changes in the sensitivity of Atlantic cod to climate-driven effects in the plankton. *Climate research* **41**, 15-19 (2010).
26. Beaugrand G, Kirby RR. Climate, plankton and cod. *Global Change Biology* **16**, 1268-1280 (2010).
27. Philips SJ, Anderson RP, Shapire RE. Maximum entropy modeling of species geographic distributions. *Ecological Modelling* **190**, 231-259 (2006).
28. Denney NH, Jennings S, Reynolds JD. Life-history correlates of maximum population growth rates in marine fishes. *Proceedings of the Royal Society B* **269**, 2229–2237 (2002).
29. Otterlei E, Folkword A, Nyhammer G, Stefansson SO. Temperature- and size-dependent growth of larval and early juvenile Atlantic cod (*Gadus morhua*): a comparative study of Norwegian coastal cod and northeast Arctic cod. *Canadian Journal of Fisheries and Aquatic Sciences* **56**, 2099-2111 (1999).
30. Caracciolo M, *et al.* Annual phytoplankton succession results from niche-environment interaction. *Journal of Plankton Research* **43**, 85-102 (2021).

31. Beaugrand G, Brander KM, Lindley JA, Souissi S, Reid PC. Plankton effect on cod recruitment in the North Sea. *Nature* **426**, 661-664 (2003).
32. Legendre P, Legendre L. *Numerical Ecology*, 2 edn. Elsevier Science B.V. (1998).
33. MacKenzie BR, Ojaveer H, Eero M. Historical ecology provides new insights for ecosystem management: eastern Baltic cod case study. *Marine Policy* **35**, 266-270 (2011).
34. Eero M, MacKenzie BR, Köster FW, Gislason H. Multi-decadal responses of a cod (*Gadus morhua*) population to human-induced trophic changes, fishing, and climate. *Ecological Applications* **21**, 214-226 (2011).
35. Pinsky ML, Jensenb OP, Ricard D, Palumbi SR. Unexpected patterns of fisheries collapse in the world's oceans. *Proceedings of the National Academy of Sciences of the United States of America* **108**, 8317–8322 (2011).
36. Andersen KH. *Fish ecology, evolution, and exploitation*. Princeton University Press (2019).
37. Engelhard GH, Righton DA, Pinnegar JK. Climate change and fishing: a century of shifting distribution in North Sea cod. *Global Change Biology* **20**, 2473-2483 (2014).
38. Brander K, Ottersen G, Wieland K, Lilly G. Decline and recovery of North Atlantic cod stocks. *GLOBEC international Newsletter* **12**, 10-12 (2006).
39. Skogen MD, Eilola K, Hansen JLS, Meier HEM, Molchanov MS, Ryabchenko VA. Eutrophication status of the North Sea, Skagerrak, Kattegat and the Baltic Sea in present and future climates: A model study. *Journal of Marine Systems* **132**, 174-184 (2014).
40. Beaugrand G. *Marine biodiversity, climatic variability and global change*. Routledge (2015).
41. ICES. ICES fisheries management reference points for category 1 and 2 stocks; Technical Guidelines. In: *ICES Advice 2021*). ICES Advisory Committee (2021).
42. Lindegren M, Mollmann C, Nielsen A, Stenseth NC. Preventing the collapse of the Baltic cod stock through an ecosystem-based management approach. *Proceedings of the National Academy of Sciences of the United States of America* **106**, 14722–14727 (2009).
43. Voss R, Quaas M. Fisheries management and tipping points: Seeking optimal management of Eastern Baltic cod under conditions of uncertainty about the future productivity regime. *Natural Resource Modeling* **35**, e12336 (2022).

44. Möllmann C, Diekmann R. Marine ecosystem regime shifts induced by climate and overfishing: A review for the Northern hemisphere. *Advances in Ecological Research* **47**, 303-347 (2012).
45. McEvoy AF. *The Fisherman's Problem: Ecology and Law in the California Fisheries, 1850-1980*. Cambridge University Press (1986).
46. O'Brien CM, Fox CJ, Planque B, Casey J. Climate variability and North Sea cod. *Nature* **404**, 142 (2000).
47. Brander K. Climate change not to blame for cod population decline. *Nature Sustainability* **1**, 262-264 (2018).
48. Cardinale M, Svedäng H. The beauty of simplicity in science: Baltic cod stock improves rapidly in a 'cod hostile' ecosystem state. *Marine Ecology Progress Series* **425**, 297-301 (2011).
49. Reid PC, *et al.* The Continuous Plankton Recorder: concepts and history, from plankton indicator to undulating recorders. *Progress in Oceanography* **58**, 117-173 (2003).
50. Munk P. Prey size spectra and prey availability of larval and small juvenile cod. *Journal of Fish Biology* **51 (Supplement A)**, 340-351 (1997).
51. Thorisson K. The food of larvae and pelagic juveniles of cod (*Gadus morhua* L.) in the coastal waters west of Iceland. *Rapport Procès verbal Réunion du Conseil International pour l'Exploration de la Mer* **191**, 264-272 (1989).
52. FAO-FIGIS. A world overview of species of interest to fisheries. Chapter: *Gadus morhua*. FIGIS Species Fact Sheets. Species Identification and Data Programme-SIDP.) (2001).
53. Cohen DM, Inada T, Iwamoto T, Scialabba N. FAO species catalogue. Vol. 10. Gadiform fishes of the world (Order Gadiformes). An annotated and illustrated catalogue of cods, hakes, grenadiers and other gadiform fishes known to date.). FAO Fish. Synop. (1990).
54. MacKenzie BR, Gislason H, Möllmann C, Köster FW. Impact of 21st century climate change on the Baltic Sea fish community and fisheries. *Global Change Biology* **13**, 1348-1367 (2007).
55. Pikitch EK, *et al.* Ecosystem-Based Fishery Management. *Science* **305**, 346-347 (2004).
56. Synnes AEW, Huserbråten M, Knutsen H, Jorde PE, Sodeland M, Moland E. Local recruitment of Atlantic cod and putative source spawning areas in a coastal seascape. *ICES Journal of Marine Science* **78**, 3767-3779 (2021).
57. Möllmann C, *et al.* Tipping point realized in cod fishery. *Scientific Reports* **11**, 14259 (2021).

58. Pörtner H-O, *et al.* Ocean systems. In: Climate Change 2014: Impacts, Adaptation, and Vulnerability. Part A: Global and Sectoral Aspects. . In: *Contribution of Working Group II to the Fifth Assessment Report of the Intergovernmental Panel on Climate Change*. (eds Field CB, *et al.*). Cambridge University Press (2014).
59. Beaugrand G, *et al.* Prediction of unprecedented biological shifts in the global ocean. *Nature Climate Change* **9**, 237-243 (2019).
60. Cooley S, *et al.* Chapter 3: Oceans and Coastal Ecosystems and their Services , In: *Climate Change 2022: Impacts, adaptation and vulnerability. Contribution of the WGII to the 6th assessment report of the intergovernmental panel on climate change. IPCC AR6 WGII*. Cambridge University Press (2022).
61. Perry AI, Low PJ, Ellis JR, Reynolds JD. Climate change and distribution shifts in marine fishes. *Science* **308**, 1912-1915 (2005).
62. Smale DA, *et al.* Marine heatwaves threaten global biodiversity and the provision of ecosystem services. *Nature Climate Change* **9**, 306-312 (2019).
63. Brander K, *et al.* Environmental impacts - Marine ecosystems. In: *North Sea region climate change assessment, regional climate studies* (eds Quante M, Colijn F). Springer Open (2016).
64. Kirby RR, Beaugrand G, Lindley JA. Synergistic effects of climate and fishing in a marine ecosystem. *Ecosystems* **12**, 548-561 (2009).